# Identifiable Token Correspondence for World Models

**Youngin Kim** [* 1]  **Ray Sun** [* 2]  **Inho Kim** [2]  **Bumsoo Park** [3]  **Hyun Oh Song** [1 2]

## Abstract

Token-based transformer world models have shown strong performance in visual reinforcement learning, but often suffer from temporal inconsistency in long-horizon rollouts, including object duplication, disappearance, and transmutation. A key reason is that most existing approaches treat next-frame prediction purely as a token generation problem, without considering the persistence of tokens across time. We introduce Identifiable Token Correspondence (ITC), a decoding step for token-based transformer world models that formulates next-frame prediction as a structured assignment problem with latent token correspondence variables: each next-frame token is explained either by copying a token from the previous frame or by generating a new one. ITC leaves the transformer architecture and training procedure unchanged and can be added on top of existing backbones. Our experiments show state-of-the-art performance on 4 challenging benchmarks. The proposed method achieves a *return* of 72.5% and a *score* of 35.6% on the Craftax-classic benchmark, significantly surpassing the previous best of 67.4% and 27.9%. We release our source code at https://github.com/snu-mllab/Identifiable-Token-Correspondence.

## 1. Introduction

Reinforcement learning (RL) provides a framework for training agents to interact with their environment through reward signals (Sutton & Barto, 2018). To avoid heavy reliance on costly environment interactions, model-based RL learns a predictive model of the environment dynamics, enabling the agent to simulate future trajectories called "imaginations"

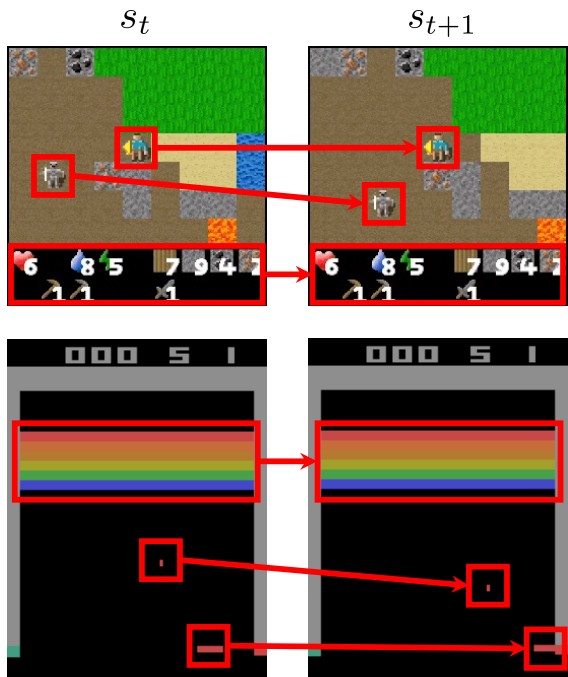

*Figure 1.* Sequential frames in visual environments like Craftax-classic and Atari contain the same underlying entities.

(Hafner et al., 2023; Micheli et al., 2022). Recently, token-based transformer world models have emerged as powerful approaches (Micheli et al., 2022; Dedieu et al., 2025). These models treat sequences of past states and actions as token streams and predict the next state token-by-token. We focus on this class of world models throughout the paper. Despite recent advances, such models often exhibit temporal inconsistency in long-horizon rollouts, including object duplication, disappearance, and transmutation into different objects. These errors compound over time and significantly limit the usefulness of long imagined trajectories for policy training.

A central reason for this failure is that most existing world models treat next-frame prediction purely as a token generation problem (Micheli et al., 2022; Dedieu et al., 2025). In realistic environments, however, many tokens in successive frames correspond to the same underlying entities that persist and move over time (see Figure 1). Predicting the next frame therefore requires determining not only what

---
[*]Equal contribution [1]Interdisciplinary Program in Artificial Intelligence, Seoul National University [2]Department of Computer Science and Engineering, Seoul National University [3]KRAFTON. Correspondence to: Hyun Oh Song <hyunoh@snu.ac.kr>.

*Proceedings of the 43rd International Conference on Machine Learning*, Seoul, South Korea. PMLR 306, 2026. Copyright 2026 by the author(s).

token should appear at each position, but also where that token comes from. When correspondence across time is not modeled explicitly, these two questions are conflated, forcing the model to relearn persistent structure at every step and making identity preservation fragile.

To address this problem, we propose Identifiable Token Correspondence (ITC), a decoding step that augments existing token-based transformer world models. ITC is not a new architecture or training recipe: it leaves the transformer and its training loss unchanged, and inserts an optimal transport solver between the transformer's next-token predictions and the final next-state tokens. The solver introduces latent variables that assign each next-frame token either to a copied token from the previous frame or to a token sampled from the transformer's predictions. This enables partial reuse of previous tokens, reducing hallucinations and improving object persistence over time, while preserving the upstream world model and its training procedure.

We evaluate ITC on the Craftax-classic, Craftax, MinAtar, and Atari 100K benchmarks. Craftax-classic is a challenging 2D open-world game featuring long-horizon tasks and dynamic enemies (Matthews et al., 2024). ITC achieves a return of 72.5% and a score of 35.6%, setting a new state-of-the-art and outperforming the previous best results of 67.4% and 27.9%, respectively (Dedieu et al., 2025). Craftax is a harder environment based on Craftax-classic, in which ITC also exceeds baselines (Matthews et al., 2024). MinAtar is a suite of 4 Atari games with simplified representations, which tests generality across different game dynamics (Young & Tian, 2019). ITC surpasses the previous state of the art for model-based RL in all 4 games (Dedieu et al., 2025). Atari 100K is a suite of 26 Atari games with diverse visual structure. ITC surpasses the previous state-of-the-art token-based world model (Cohen et al., 2025) across the 26 games.

## 2. Preliminaries

### 2.1. Model-based Reinforcement Learning

Reinforcement learning considers a Partially Observable Markov Decision Process (POMDP), characterized by $(\mathbb{S}, \mathbb{A}, \Omega, T, O, R, \gamma)$, where $\mathbb{S}$ is a set of states, $\mathbb{A}$ is a set of discrete actions, $\Omega$ is a set of observations, $T$ gives the transition probabilities between states $T(s' \mid s, a)$, $O$ gives the observation probabilities $O(o \mid s)$, and $R$ is a reward function $R(s, a)$ (Sutton & Barto, 2018). The goal is to find a policy $\pi$ which chooses actions for each state that maximizes the expected discounted return $\mathbb{E}_\pi \left[ \sum_{t \geq 0} \gamma^t r_t \right]$, where $\gamma$ is a discount factor. A world model takes an input of previous state $s_t$ and action $a_t$, then returns a predicted output of next state $\hat{s}_{t+1}$, reward $r_t$, and done signal $d_t$, similar to the real environment. The agent collects real environment trajectories during training by interacting with

the environment using the policy $\pi$. Then the world model trains on the trajectories saved in the replay buffer. Over the course of training, the agent is trained on both the trajectories collected from the real environment and generated trajectories from the world model, called *imaginations*.

### 2.2. RoPE

Rotary Position Embedding (RoPE) is a positional encoding method that injects positional information into a transformer's attention mechanism by applying rotations to query and key vectors (Su et al., 2024). These rotations cause the attention operation to naturally encode relative offsets between tokens. Concretely, each input token embedding is partitioned into pairs of coordinates, with each pair forming a 2D subspace where a rotation is applied according to the token's 1D position index. Owing to its simplicity and scalability, RoPE has become the standard positional encoding in modern transformer architectures.

However, RoPE uses a single-dimensional position index, which is unable to distinguish between temporal differences (i.e., tokens from different time steps) and spatial differences (i.e., tokens from different positions within the same frame). To incorporate both spatial and temporal information into the model, 3D positional encoding for multi-dimensional information has been developed (Wang et al., 2024; Wei et al., 2025). Each token's embedding is divided into three sub-vectors corresponding to its temporal, vertical, and horizontal coordinates. RoPE is then applied independently along each axis, enabling the attention mechanism to capture localized relational structure across both space and time. This formulation allows the model to generalize over local interactions (e.g., neighboring pixels or frames), regardless of absolute location. It preserves adjacency in both spatial and temporal dimensions, while original RoPE loses the adjacency of the vertical axis and temporal axis. On top of 3D RoPE, adding absolute positional embeddings also improves token representations (Agarwal et al., 2025).

### 2.3. Tokenizer

Transformer world models require a tokenizer to convert states and actions into discrete tokens for the transformer. The state tokenizer maps each frame to a sequence of $L$ tokens $\{q_1, \ldots, q_L\}$, where each $q_i \in \{1, \ldots, K\}$ indexes one of $K$ codebook entries, and reconstructs a frame from such a sequence.

Dedieu et al. (2025) introduced a tokenizer based on nearest neighbor patch lookup, where each token corresponds to a fixed visual patch. Each frame is divided into a grid of $L$ patches $\{p_1, \ldots, p_L\}$ with $p_i \in [0, 1]^{h \times w \times 3}$, and the codebook $C = \{c_1, \ldots, c_K\}$ consists of $K$ patches $c_k \in [0, 1]^{h \times w \times 3}$. Each patch is mapped to a token by nearest

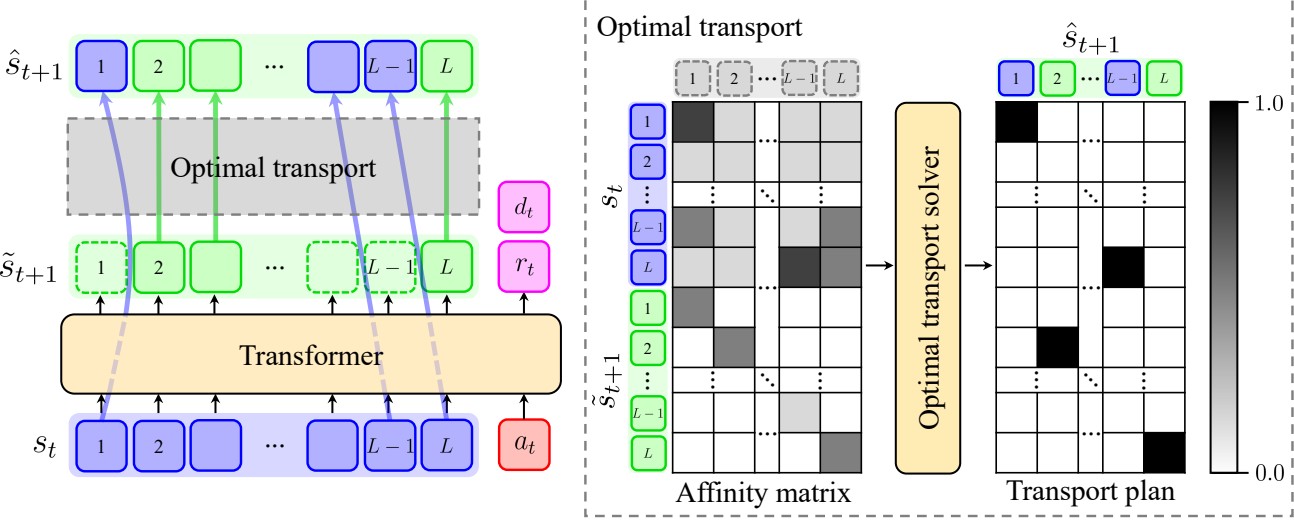

*Figure 2.* Our proposed world model enhances next state prediction by solving an optimal transport problem with previous state tokens ($s_t$, blue) and the transformer's output for candidate next-state tokens ($\tilde{s}_{t+1}$, green) to generate the final next-state tokens ($\hat{s}_{t+1}$). Optimal transport defines an affinity matrix from the $s_t$ and $\tilde{s}_{t+1}$ tokens to the positions for $\hat{s}_{t+1}$. A solver takes the affinity matrix and produces a transport plan, assigning a token from $s_t$ or $\tilde{s}_{t+1}$ to each final next-state token in $\hat{s}_{t+1}$. This approach enables effective reuse of relevant past tokens.

neighbor lookup:

$$q_i = \text{enc}(p_i) = \operatorname*{argmin}_{1 \leq k \leq K} \|p_i - c_k\|_2^2.$$

The codebook is constructed by sampling patches from the replay buffer. A patch is added if it is sufficiently far from all existing codes: when $\min_{1 \leq k \leq K} \|p_i - c_k\|_2^2 > \tau$ for a chosen threshold $\tau$. To reconstruct the frame, each token is mapped back to its code, $\text{dec}(q_i) = c_{q_i}$, and the patches are reassembled into the full image.

Cohen et al. (2025) trains a VQ-VAE tokenizer to reconstruct the full frame (Van Den Oord et al., 2017). Instead of operating on raw patches, a convolutional encoder $f_{\text{enc}}$ maps each frame to a grid of $L$ continuous feature vectors $\{z_1, \ldots, z_L\}$ with $z_i \in \mathbb{R}^d$, and the codebook $E = \{e_1, \ldots, e_K\}$ consists of $K$ learnable embeddings $e_k \in \mathbb{R}^d$. Each feature is mapped to a token by nearest neighbor lookup in the codebook:

$$q_i = \text{enc}(z_i) = \operatorname*{argmin}_{1 \leq k \leq K} \|z_i - e_k\|_2^2.$$

A convolutional decoder $f_{\text{dec}}$ then reconstructs the frame from the embeddings $\{e_{q_1}, \ldots, e_{q_L}\}$ of the selected codes. The encoder, decoder, and codebook are trained end-to-end with reconstruction and commitment losses.

### 2.4. Optimal Transport

Optimal transport is a family of optimization problems that compares and aligns probability distributions based on a given cost of moving mass between elements (Peyré & Cuturi, 2019). Optimal transport considers probability distributions $\mathbf{a} \in \Delta^{n-1}$ and $\mathbf{b} \in \Delta^{m-1}$ over the source and target domains, respectively. Given a cost matrix $C$, it seeks a transport plan $\mathbf{\Pi} \in \mathbb{R}_+^{n \times m}$ that minimizes the cost $\langle \mathbf{\Pi}, C \rangle = \sum_{i=1}^{n} \sum_{j=1}^{m} \Pi_{ij} C_{ij}$, subject to the marginal constraints $\mathbf{\Pi} \mathbf{1}_m = \mathbf{a}$ and $\mathbf{\Pi}^\top \mathbf{1}_n = \mathbf{b}$.

To solve optimal transport problems efficiently, regularized variants of optimal transport have been proposed. One popular approach introduces an entropic regularization term to the objective, leading to the *Sinkhorn distance*, which can be computed efficiently using iterative matrix scaling (Cuturi, 2013). The Sinkhorn algorithm solves the regularized problem in $O(n^2/\epsilon^2)$ time for a desired approximation error $\epsilon$, making it practical for large-scale problems.

## 3. Method

Based on the concepts presented in Section 2, ITC augments a token-based transformer world model with an optimal-transport decoding step that models token correspondence between frames. ITC is not a new architecture or training procedure: it leaves the transformer and its training loss unchanged, and inserts an optimal transport solver between the transformer's next-token predictions and the final next-state tokens. After the tokenizer converts states and actions to tokens, the token embeddings are augmented with 3D positional encodings adopted from prior work (Wei et al., 2025; Agarwal et al., 2025), before being fed into a transformer. The transformer output tokens are then used by the optimal

**Algorithm 1** Decoding with Optimal Transport

> **Input:** transformer prediction $\mathbf{p}$,
> previous tokens $\mathbf{u}$,
> number of tokens per frame $L$,
> Sinkhorn regularization parameter $\epsilon$,
> Number of Sinkhorn iterations $T$
> **Output:** Generated tokens for next frame $\mathbf{u}'$
>
> Compute $\boldsymbol{A}^{(prev)}$, $\boldsymbol{A}^{(gen)}$ from Equations 1 and 2
>
> $$\boldsymbol{A} = \begin{pmatrix} \boldsymbol{A}^{(prev)} & \boldsymbol{0} \in \mathbb{R}^{L \times L} \\ \boldsymbol{A}^{(gen)} & \boldsymbol{0} \in \mathbb{R}^{L \times L} \end{pmatrix} \in \mathbb{R}^{(2L) \times (2L)}$$
>
> $\boldsymbol{P} = \text{SINKHORN}(-\boldsymbol{A}, \epsilon, T)$
> $\boldsymbol{P}^{(prev)} = \boldsymbol{P}[1:L, 1:L]$
> $\boldsymbol{P}^{(gen)} = \boldsymbol{P}[L+1:2L, 1:L]$
>
> $\boldsymbol{\Pi}^{(prev)}, \boldsymbol{\Pi}^{(gen)} = \text{BINARIZATION}(\boldsymbol{P}^{(prev)}, \boldsymbol{P}^{(gen)})$
>
> **for** $j = 0$ to $L - 1$ **do**
>   **if** $\Pi_{ij}^{(prev)} = 1$ for some $i$ **then**
>     $\mathbf{u}'_j = \mathbf{u}_i$
>   **else if** $\Pi_{jj}^{(gen)} = 1$ **then**
>     $\mathbf{u}'_j = \text{sample}(\mathbf{p}_j)$
>   **end if**
> **end for**
> **Return** $\mathbf{u}'$

**Algorithm 2** BINARIZATION of partial transport plan

> **Input:** Partial transport plans $\boldsymbol{P}^{(prev)}$, $\boldsymbol{P}^{(gen)}$,
> large value $v$
> **Output:** Binarized transport plans $\boldsymbol{\Pi}^{(prev)}$, $\boldsymbol{\Pi}^{(gen)}$
>
> $\boldsymbol{P}^{\text{in}} = \text{concatenate}\left(\boldsymbol{P}^{(prev)}, \boldsymbol{P}^{(gen)}\right)$
> Initialize $\boldsymbol{P}^{(0)} = \boldsymbol{P}^{\text{in}}$, $t = 0$
> **repeat**
>   target = $\text{argmax}(\boldsymbol{P}^{(t)}, \dim = 1)$
>
>   $\boldsymbol{\Pi}^{\text{initial}} = \boldsymbol{0}_{n \times m}$
>   **for** $i = 0$ to $n - 1$ **do**
>     $\boldsymbol{\Pi}^{\text{initial}}[i, \text{target}[i]] = 1$
>   **end for**
>
>   $\boldsymbol{C} = \boldsymbol{P}^{(t)} \odot \boldsymbol{\Pi}^{\text{initial}} - v(1 - \boldsymbol{\Pi}^{\text{initial}})$
>   source = $\text{argmax}(\boldsymbol{C}, \dim = 0)$
>   $\boldsymbol{\Pi}^{\text{out}} = \boldsymbol{0}_{n \times m}$
>   **for** $j = 0$ to $m - 1$ **do**
>     $\boldsymbol{\Pi}^{\text{out}}[\text{source}[j], j] = 1$
>   **end for**
>
>   $\boldsymbol{\Pi}^{\text{out}} = \boldsymbol{\Pi}^{\text{out}} \odot \boldsymbol{\Pi}^{\text{initial}}$
>   $\boldsymbol{R} = (1 - \boldsymbol{\Pi}^{\text{out}}) \odot \boldsymbol{\Pi}^{\text{initial}}$
>   $\boldsymbol{P}^{(t+1)} = \boldsymbol{P}^{(t)} - v\boldsymbol{R}$
>   $t = t + 1$
> **until** $\boldsymbol{\Pi}^{\text{out}} = \boldsymbol{\Pi}^{\text{initial}}$
>
> $\boldsymbol{\Pi}^{(prev)} = \boldsymbol{\Pi}^{\text{out}}[1:L, 1:L]$
> $\boldsymbol{\Pi}^{(gen)} = \boldsymbol{\Pi}^{\text{out}}[L+1:2L, 1:L]$
> **Return** $\boldsymbol{\Pi}^{(prev)}, \boldsymbol{\Pi}^{(gen)}$

transport solver to produce the next state tokens, as shown in Figure 2. Because ITC operates on the discrete tokens produced by an arbitrary frame tokenizer, the same decoding step applies whether tokens come from nearest-neighbor patch lookup or from a learned VQ-VAE; we exercise both choices in our experiments.

Our transformer world model uses optimal transport as the identifiable token correspondence mechanism. In existing approaches, the output of the transformer world model is directly used to predict each token in the next frame (Micheli et al., 2022; 2024; Agarwal et al., 2024; Dedieu et al., 2025). However, in most visual environments, two adjacent frames are often very similar, e.g. the same tiles but shifted when the player moves right. Our intuition is closely related to the notion of optical flow from classic computer vision tasks (Brox et al., 2004; Vedula et al., 2005; Perazzi et al., 2016). This relation allows tokens to be taken directly from the previous frame into the next frame, rather than offloading the burden of regenerating all next-state tokens to the transformer. To exploit this, the final next-state token predictions are formulated as an optimal transport problem. The end-to-end decoding process is characterized in Algorithm 1.

Let $L$ be the number of tokens for each frame state. Our method constructs a graph $\mathcal{G} = (\mathcal{V}, \mathcal{E})$, where the vertices

$\mathcal{V} = \mathcal{V}_S \cup \mathcal{V}_D$ consist of source vertices $\mathcal{V}_S$ that correspond to previous state tokens and candidate next-state tokens, and destination vertices $\mathcal{V}_D$ that represent the finalized next-state tokens ($|\mathcal{V}_S| = 2L$ and $|\mathcal{V}_D| = L$). The edges $\mathcal{E} = \{(u, v) \mid u \in \mathcal{V}_S, v \in \mathcal{V}_D\}$ connect all sources to all destinations. We now define affinities on these edges for transport.

Let $K$ be the size of the codebook. Given transformer predictions $\mathbf{p}_j \in [0, 1]^K$ for the next state tokens, and previous state tokens $\mathbf{u}_i \in \{0, 1\}^K$ for all $i, j \in \{0, \ldots, L - 1\}$, we define an affinity matrix $\boldsymbol{A}^{(prev)} \in \mathbb{R}^{L \times L}$ that scores the affinity between previous state tokens and predicted next-state tokens. Each entry is computed as:

$$A_{ij}^{(prev)} = \langle \mathbf{p}_j, \mathbf{u}_i \rangle - c_d D\left((x_i, y_i), (x_j, y_j)\right), \quad (1)$$
$$\forall i, j \in \{0, \ldots, L - 1\},$$

where $c_d$ is a coefficient of cost for distance, $D(\cdot)$ is a distance function for 2D coordinates, and $(x_i, y_i)$ and $(x_j, y_j)$ are the 2D coordinates of the $i$-th and $j$-th tokens, respectively. In practice, $D$ is a simple cap on inter-frame token

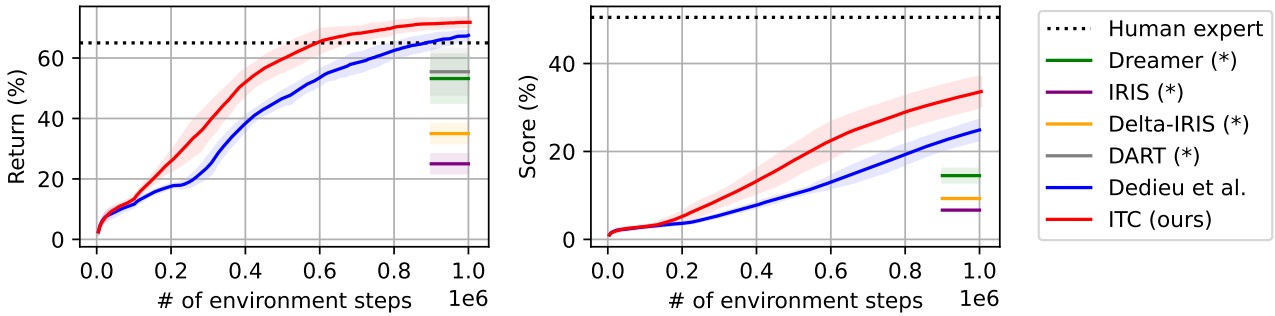

*Figure 3.* ITC achieves state-of-the-art return and score in Craftax-classic, with significantly faster convergence (Matthews et al., 2024). Shading indicates standard deviation among seeds. *Baselines with reported results at 1M steps are displayed with horizontal lines from 900K to 1M steps. DART does not report score, and IRIS and $\Delta$-IRIS do not report standard deviation for score.

displacement (Appendix B); we use the same cap across all four benchmarks, so applying ITC to a new environment does not require any environment-specific knowledge. To allow the model to generate new content not present in the previous frame, the graph includes wildcard tokens. The matrix $\boldsymbol{A}^{(gen)} \in \mathbb{R}^{L \times L}$ scores the bonus of admitting newly generated tokens instead of reusing the previous ones, using diagonal entries:

$$A_{kj}^{(gen)} = \begin{cases} \|\mathbf{p}_j\|_\infty - c_w, & \text{if } k = j, \\ -\infty & \text{otherwise,} \end{cases} \quad (2)$$
$$\forall k, j \in \{0, \ldots, L-1\},$$

where $c_w$ is a constant penalty for using a wildcard token. With the matrices defined above, an optimal transport plan $\boldsymbol{P}^{(prev)}$ and $\boldsymbol{P}^{(gen)}$ is computed by optimizing the following equation:

$$\underset{\substack{\boldsymbol{P}^{(prev)} \in [0,1]^{L \times L} \\ \boldsymbol{P}^{(gen)} \in [0,1]^{L \times L}}}{\text{minimize}} \left\langle -\begin{pmatrix} \boldsymbol{A}^{(prev)} \\ \boldsymbol{A}^{(gen)} \end{pmatrix}, \begin{pmatrix} \boldsymbol{P}^{(prev)} \\ \boldsymbol{P}^{(gen)} \end{pmatrix} \right\rangle$$

$$\text{subject to} \quad \boldsymbol{P}^{(prev)} \mathbf{1}_L \le \mathbf{1}_L, \quad (3)$$
$$\boldsymbol{P}^{(gen)} \mathbf{1}_L \le \mathbf{1}_L,$$
$$\left( \boldsymbol{P}^{(prev)} + \boldsymbol{P}^{(gen)} \right)^\top \mathbf{1}_L = \mathbf{1}_L.$$

Solving this optimization problem involves the Sinkhorn algorithm. By default, the Sinkhorn algorithm minimizes the objective given by a cost matrix rather than an affinity matrix, so the cost matrix is set as the negative of the computed affinity matrix. Solving the optimal transport problem yields a partial transport plan, represented by a matrix with continuous values in the range $[0, 1]$.

However, our application requires a strict one-to-one mapping between discrete tokens. To address this, we convert the partial transport plan into a binary assignment matrix

with values $\{0, 1\}$ using a greedy binarization procedure based on column-wise argmax. Specifically, for each column in the transport matrices $\boldsymbol{P}^{(prev)}$ and $\boldsymbol{P}^{(gen)}$, we identify the row with the highest transport weight, selecting that row in either $\boldsymbol{P}^{(prev)}$ or $\boldsymbol{P}^{(gen)}$, whichever yields the larger value. In the event of a conflict where multiple columns select the same row, we retain the assignment corresponding to the column with the higher transport value and reassign the conflicting column using argmax again, excluding rows that have already been assigned. The complete binarization procedure, which is adapted from Kim et al. (2020), is described in Algorithm 2.

Let $\boldsymbol{\Pi}^{(prev)} \in \{0, 1\}^{L \times L}$ and $\boldsymbol{\Pi}^{(gen)} \in \{0, 1\}^{L \times L}$ denote the binarized versions of $\boldsymbol{P}^{(prev)}$ and $\boldsymbol{P}^{(gen)}$, respectively. $\boldsymbol{\Pi}^{(prev)}$ and $\boldsymbol{\Pi}^{(gen)}$ are the latent variables that represent token correspondence. The $j$-th token of the next state is determined by copying the $i$-th token of the previous state where $\Pi_{ij}^{(prev)} = 1$. If no such $i$ exists, which occurs only when $\Pi_{jj}^{(gen)} = 1$, the model instead samples from the transformer's predicted distribution. The overall decoding rule is thus defined as

$$\mathbf{u}_j' = \begin{cases} \mathbf{u}_i, & \text{where } \Pi_{ij}^{(prev)} = 1, \\ \text{sample}(\mathbf{p}_j) & \text{where } \Pi_{jj}^{(gen)} = 1, \end{cases} \quad (4)$$
$$\forall j \in \{0, \ldots, L-1\}.$$

By using the latent correspondence variables $\boldsymbol{\Pi}^{(prev)}$ and $\boldsymbol{\Pi}^{(gen)}$ in this way, the world model selectively reuses tokens that have a strong correspondence with the previous frame, while leveraging the transformer to generate new tokens for changes in the environment.

*Table 1.* Results on Craftax-classic after 0.5M and 1M environment interactions. Our *direct baseline* is the world model of Dedieu et al. (2025); ITC adds an optimal-transport decoding step on top of this same backbone without modifying its architecture or training. Rows labeled "(reproduced)" are run with our code, and the row marked † uses ITC's hyperparameters for a head-to-head comparison. Return is averaged over episodes of the final 50,000 environment interactions to smooth out variance. Score is reported directly, as it is already a cumulative metric. Metrics not reported by baselines are marked as —.

| | @ 0.5M | | @ 1M | |
| --- | --- | --- | --- | --- |
| Method | Return (%) | Score (%) | Return (%) | Score (%) |
| Human expert | — | — | $65.0 \pm 10.5$ | $50.5 \pm 6.8$ |
| DreamerV3 (Hafner et al., 2023) | — | — | $53.2 \pm 8.0$ | $14.5 \pm 1.6$ |
| IRIS (Micheli et al., 2022) | — | — | $25.0 \pm 3.2$ | 6.66 |
| $\Delta$-IRIS (Micheli et al., 2024) | — | — | $35.0 \pm 3.2$ | 9.30 |
| DART (Agarwal et al., 2024) | — | — | $55.45 \pm 3.39$ | — |
| Dedieu et al. (2025) | — | — | $67.42 \pm 0.55$ | $27.91 \pm 0.63$ |
| Dedieu et al. (2025) (reproduced) | $48.17 \pm 0.82$ | $10.22 \pm 0.20$ | $68.14 \pm 0.42$ | $24.89 \pm 0.74$ |
| Dedieu et al. (2025) (reproduced)† | $54.32 \pm 0.60$ | $13.06 \pm 0.39$ | $68.55 \pm 0.72$ | $27.24 \pm 0.86$ |
| ITC (ours) | $\mathbf{63.10} \pm 1.24$ | $\mathbf{20.12} \pm 0.80$ | $\mathbf{72.46} \pm 0.45$ | $\mathbf{35.60} \pm 0.92$ |

## 4. Experiments

### 4.1. Craftax-classic

**Environment** We evaluate our method on the Craftax-classic environment (Matthews et al., 2024). Craftax-classic is a fast implementation of Crafter, a challenging procedurally generated, partially observable environment featuring stochastic transitions and a complex hierarchy of achievements (Hafner, 2021). These attributes demand both strong generalization and the ability to model object interactions across time.

**Experiment Configuration** Each method is trained on Craftax-classic for 1M environment steps, using 10 different seeds per method. The baseline methods consist of DreamerV3 (Hafner et al., 2023), IRIS (Micheli et al., 2022), $\Delta$-IRIS (Micheli et al., 2024), DART (Agarwal et al., 2024), and Dedieu et al. (2025)[1], which had the previous state-of-the-art return on Craftax-classic. Each experiment runs on a single Nvidia RTX 3090 GPU for 48.2 hours. See Appendix A for all hyperparameters.

**Results** The direct baseline for ITC on Craftax-classic is Dedieu et al. (2025), the previous state-of-the-art. ITC keeps this backbone, including its patch tokenizer, transformer architecture, and training procedure. On top of improving the RoPE implementation to use 3D RoPE from Section 2.2, ITC replaces the transformer decoding with the optimal-transport decoding step from Section 3. Figure 3 shows that adding ITC on top of this same backbone leads to substan-

tially higher return and score, along with faster convergence, compared to all baselines.[2] Return and score are reported in Table 1, as the mean and standard error over 10 seeds. After 1M environment interactions, ITC surpasses all baselines on both metrics. ITC also outperforms the previous best baseline at 0.5M environment interactions, demonstrating superior sample efficiency in a more data-constrained setting. Table 2 disentangles the gains of the baseline improved with 3D RoPE vs. the optimal transport decoding step. ITC still shows a statistically significant gap compared to the improved baseline and more consistent improvement.

*Table 2.* Component ablation on Craftax-classic after 1M environment interactions, with each row adding one component to the baseline backbone. 3D RoPE refers to the adopted 3D spatio-temporal positional encoding (Wei et al., 2025; Agarwal et al., 2025). † uses ITC's hyperparameters.

| Configuration | Return (%) | Score (%) |
| --- | --- | --- |
| Dedieu et al. (2025)† | $68.55 \pm 0.72$ | $27.24 \pm 0.86$ |
| + 3D RoPE | $71.85 \pm 0.63$ | $33.94 \pm 1.10$ |
| + ITC | $\mathbf{72.46} \pm 0.45$ | $\mathbf{35.60} \pm 0.92$ |

**Accuracy Evaluation** To directly assess the contribution of ITC to world model prediction, we measure how often the next state is predicted exactly (i.e., every token is correct) on 10,000 held-out Craftax-classic transitions. Table 3 shows that adding ITC to the same transformer backbone raises

---

[1]We use the (fast) variant from Dedieu et al. (2025), as the (slow) variant is prohibitively expensive to train.

[2]Score is a metric defined as the geometric mean of the success rates for each achievement (Hafner, 2021). Score puts more emphasis on unlocking a variety of achievements, in contrast to return, which is simply the sum of rewards for each episode.

*Table 3.* Per-step prediction accuracy on 10,000 held-out Craftax-classic transitions. *Overall* is the fraction of next states for which every token is predicted correctly; the other two columns split this set by whether the input state contains a randomly moving creature. Adding ITC on top of the same transformer backbone lifts overall per-step accuracy, with the largest gain on the harder creature-containing transitions. † uses hyperparameters of ITC.

| Method | Overall accuracy (%) | Accuracy with creatures (%) | Accuracy without creatures (%) |
|---|---|---|---|
| Dedieu et al. (2025)† | 46.94 | 33.83 | 61.03 |
| ITC (ours) | **51.13** | **38.37** | **65.46** |

overall per-step accuracy. The gain is largest on the hardest subset, transitions containing a randomly moving creature, and the improvement is also consistent on the easier subset without creatures. At the finer token level, ITC reduces the per-token prediction error rate from $6.76\%$ for Dedieu et al. (2025) to $5.79\%$. Because imagination rollouts unroll the world model over many steps, per-step errors compound across time. ITC's higher per-step accuracy therefore translates directly into the improved returns and scores reported in Table 1, providing direct evidence that its downstream policy gains come from a more accurate world model.

**Qualitative Analysis** Figure 4 compares imaginations generated by our method vs. Dedieu et al. (2025). Our method excels in situations where tiles in the generated frame are correlated. For example, a creature in Craftax-classic can move to adjacent tiles, but it should only move to one tile and should not be duplicated to multiple tiles. However, because the transformer generates output tokens for a state in parallel, it cannot capture this constraint naturally. Therefore, during imagination, duplication or disappearance of creatures occurs, which is a critical defect of modeling environment dynamics. ITC eliminates this issue by capturing the appropriate constraint between output tiles. Solving this issue is particularly important because similar hallucinations arise in non-transformer world models as well. For example, Figure 5 shows duplication and disappearance artifacts in an imagination rollout generated by DreamerV3 (Hafner et al., 2023). Thus, by eliminating these hallucinations, ITC resolves a problem that is widespread among world models.

**Compute Time Analysis** Table 4 reports the running time of ITC, and its baseline Dedieu et al. (2025) using the same hyperparameters. ITC increases the overall end-to-end training time by only $2.8\%$. Thus, ITC introduces negligible overhead to world model and policy training.

**Hyperparameter Sensitivity** ITC introduces two hyperparameters: a distance cost coefficient $c_d$ and a wildcard penalty $c_w$. Table 5 sweeps each around the chosen value on Craftax-classic. Both return and score remain within a narrow band across all settings, indicating that ITC is robust to the choice of these hyperparameters and does not require extensive tuning to perform well.

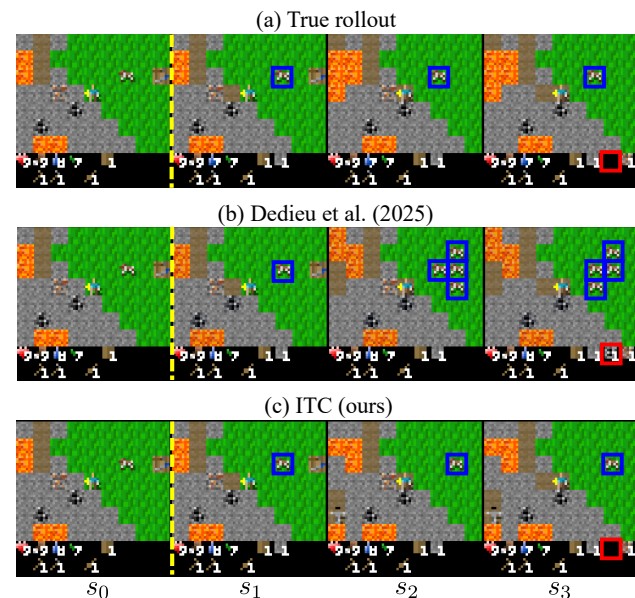

*Figure 4.* Comparison of imagined rollouts from different world models. (a) shows the ground-truth environment trajectory, while (b) and (c) illustrate imagined rollouts generated by the baseline and ITC, respectively. All rollouts begin from the same initial state $s_0$ (left of the yellow dashed line). ITC fixes inaccurate dynamics (red boxes) and duplication errors (blue boxes) produced by the baseline.

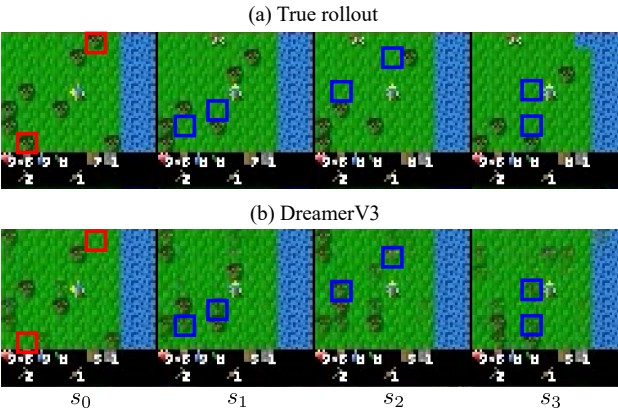

*Figure 5.* An imagination rollout of DreamerV3 compared to the ground-truth trajectory. DreamerV3's imagination includes disappearance of trees (red boxes) and duplication of trees (blue boxes) over time, similar to duplication issues shown in Figure 4 for Dedieu et al. (2025).

*Table 4.* Running times on a single Nvidia RTX 3090 GPU. WM training measures one epoch of world model training. Imagination measures one epoch of policy training in imagination. Total time represents end-to-end training time for 1M environment steps. † uses hyperparameters of ITC.

| Method | WM training (minutes) | Imagination (minutes) | Total time (hours) |
|---|---|---|---|
| Dedieu et al. (2025)[†] | 8.08 | 4.48 | 46.3 |
| ITC (ours) | 8.19 | 4.78 | 48.2 |

*Table 5.* Sensitivity of ITC to its optimal transport hyperparameters on Craftax-classic. ITC is robust to a wide range of values for both the distance cost coefficient $c_d$ and the wildcard penalty $c_w$.

*(a)* Sweep over $c_d$, with $c_w = 0.3$ fixed.

| $c_d$ | Return (%) | Score (%) |
|---|---|---|
| 0.0 | 72.08 | 32.89 |
| 0.3 | 71.10 | 31.41 |
| 0.6 | 72.46 | 35.60 |
| 0.8 | 71.49 | 33.54 |

*(b)* Sweep over $c_w$, with $c_d = 0.6$ fixed.

| $c_w$ | Return (%) | Score (%) |
|---|---|---|
| 0.3 | 72.46 | 35.60 |
| 0.6 | 71.60 | 35.41 |

*Table 6.* Results on Craftax after 1M environment interactions. ITC is built on the world model of Dedieu et al. (2025) (the direct baseline, top row); Simulus is included for reference (it does not report Score, —).

| Method | Return (%) | Score (%) |
|---|---|---|
| Dedieu et al. (2025) | 5.44 ± 0.25 | 1.53 ± 0.10 |
| Simulus | 6.59 | — |
| ITC (ours) | **7.09** ± 0.20 | **2.40** ± 0.04 |

## 4.2. Craftax

Craftax is a more complex and difficult environment that builds on Craftax-classic (Matthews et al., 2024). Craftax features a larger screen, more items, more enemies, and more levels compared to Craftax-classic. As in Craftax-classic, our direct baseline is the world model of Dedieu et al. (2025)[3], on top of which ITC adds the optimal-transport decoding step; we additionally include Simulus (Cohen et al., 2025) for reference since it held the previous best return on Craftax. All hyperparameters are listed in Appendix C. Table 6 reports return and score on Craftax, as the mean and standard error over 5 seeds. ITC achieves a return of 7.09% and a score of 2.40%, surpassing both baselines. These results demonstrate that ITC's decoding step continues to help in a more difficult environment.

## 4.3. MinAtar

To further validate the generalization performance of our approach, we also evaluate on the MinAtar benchmark (Young & Tian, 2019; Lange, 2022). MinAtar consists of 4 Atari games with simplified symbolic observations of size $10 \times 10$. As in Craftax-classic, our *direct baseline* is the world model

---

[3]We report the (fast) variant from version arXiv:2502.01591v1 of Dedieu et al. (2025).

of Dedieu et al. (2025), which previously held state-of-the-art for model-based RL on MinAtar; ITC reuses this backbone and only adds the optimal-transport decoding step from Section 3. We additionally include the recent model-free Artificial Dopamine (AD) agent (Guan et al., 2023) for reference. Each method is trained on each game in MinAtar for 1M environment steps (except AD uses 5M steps), using 10 seeds per game. Table 7 shows that ITC outperforms both baselines in all 4 games; the gap to Dedieu et al. (2025) isolates the gain from ITC's decoding step under an otherwise identical world model. Return graphs for each game can be found in Appendix D. By improving in every game, ITC demonstrates robust benefits across a variety of environments.

## 4.4. Atari 100K

We further assess performance on the popular Atari 100K benchmark, which trains on a suite of 26 Atari games for 100K environment interactions each, to validate the generalization of our method to non-grid environments (Kaiser et al., 2020). It includes various visual dynamics, such as teleportation, room changes, and first-person point of view. We use the current state-of-the-art token-based world model, Simulus (Cohen et al., 2025), as our baseline, since the Dedieu et al. (2025) baseline is not designed for or tested on Atari 100K. Unlike the patch-lookup tokenizer used in Craftax-classic, Craftax, and MinAtar, Simulus uses a learned VQ-VAE tokenizer (Section 2.3). We create an instantiation of ITC for Atari 100K by applying our method to Simulus without modification, demonstrating that ITC works directly on tokens from a different tokenizer family. Each method is trained on each game in Atari 100K for

*Table 7.* Returns on MinAtar after 1M environment interactions (or 5M for AD). ITC is built on the world model of Dedieu et al. (2025) (the previous model-based state of the art); the row immediately above ITC is the direct baseline, so the ITC vs. Dedieu et al. (2025) gap isolates the contribution of the optimal-transport decoding step. Return is evaluated on 1,000 evaluation episodes at the end of training.

| Method | Asterix | Breakout | Freeway | SpaceInvaders |
|---|---|---|---|---|
| AD (Guan et al., 2023) | $21.05 \pm 0.65$ | $27.78 \pm 0.16$ | $57.68 \pm 0.07$ | $140.36 \pm 1.70$ |
| Dedieu et al. (2025) | $44.81 \pm 3.54$ | $93.92 \pm 1.44$ | $71.12 \pm 0.13$ | $186.16 \pm 1.25$ |
| ITC (ours) | $\mathbf{50.04} \pm 2.98$ | $\mathbf{99.53} \pm 2.31$ | $\mathbf{71.34} \pm 0.07$ | $\mathbf{188.85} \pm 0.62$ |

*Table 8.* Aggregate metrics on Atari 100K after 100K environment interactions. ITC is built on top of Simulus (Cohen et al., 2025), the previous state-of-the-art token-based world model on this benchmark, so the Simulus vs. ITC gap isolates the contribution of the optimal-transport decoding step on a VQ-VAE-based backbone. Return for each game is evaluated on 100 evaluation episodes at the end of training.

| Method | IQM ($\uparrow$) | Optimality Gap ($\downarrow$) |
|---|---|---|
| DreamerV3 (Hafner et al., 2023) | 0.487 | 0.510 |
| STORM (Zhang et al., 2023) | 0.561 | 0.472 |
| Diamond (Alonso et al., 2024) | 0.641 | 0.480 |
| Simulus (Cohen et al., 2025) | 0.990 | 0.412 |
| ITC (ours) | **1.092** | **0.376** |

100K environment steps, using 5 seeds per game. Results on Atari 100K are reported as human-normalized score, calculated as $\frac{\text{agent\_return} - \text{random\_agent\_return}}{\text{human\_return} - \text{random\_agent\_return}}$. Table 8 shows that ITC exceeds the baseline and achieves new state-of-the-art performance in interquartile mean (IQM) and optimality gap, the robust metrics proposed by Agarwal et al. (2020). Detailed results for each game are presented in Table 13 of Appendix E. By excelling in Atari 100K, ITC shows that its performance generalizes across 2D visual RL environments.

## 5. Related Work

### 5.1. Transformer World Models

Transformer architectures have been effectively utilized in model-based RL. The concept of transformer world models was first introduced by IRIS (Micheli et al., 2022). Building upon IRIS, $\Delta$-IRIS proposed an agent architecture that encodes stochastic deltas between time steps, enhancing token efficiency by exploiting similarities between adjacent frames (Micheli et al., 2024). TWM, STORM, DART, and TWISTER also incorporated transformer world models, demonstrating their efficacy across different benchmarks (Robine et al., 2023; Zhang et al., 2023; Agarwal et al., 2024; Burchi & Timofte, 2025). Transformer world models further advanced with techniques including nearest neighbor tokenization and block teacher forcing, achieving state-of-the-art performance on Craftax-classic (Dedieu et al., 2025). Outside of transformers, other world models have used GRUs (Hafner et al., 2023), diffusion (Alonso et al.,

2024), decoder-free latent spaces (Hansen et al., 2024), and discrete codebook latent spaces (Scannell et al., 2025).

### 5.2. Optimal Transport in RL

Optimal transport theory has been applied to RL in other contexts, specifically for curriculum and offline reinforcement learning. CurrOT framed curriculum generation as a constrained optimal transport problem between task distributions (Klink et al., 2022). GRADIENT formulated curriculum reinforcement learning as an optimal transport problem with a tailored distance metric between tasks (Huang et al., 2022). Additionally, Achievement Distillation introduced a contrastive learning method using optimal transport to enhance the discovery of hierarchical achievements, leading to improved sample efficiency (Moon et al., 2023).

## 6. Conclusion

In this paper, we present ITC, a transformer world model that captures token correspondences between frames using optimal transport. ITC identifies the underlying entities inherent in visual environments, preventing temporal inconsistency such as duplicated or disappearing objects. By selectively reusing tokens from preceding frames, it effectively leverages frame-to-frame similarities to model next-state tokens instead of solely relying on the transformer to regenerate each one. This enables ITC to achieve new state-of-the-art performance on the challenging Craftax-classic, Craftax, MinAtar, and Atari 100K benchmarks.

## Acknowledgements

This work was supported by the Air Force Office of Scientific Research under award number FA2386-25-1-4013, a grant from KRAFTON AI, an Institute of Information & Communications Technology Planning & Evaluation (IITP) grant funded by the Korea government (MSIT) [No. RS-2026-25524173, Ultra-Long-Term Hierarchical Memory and Reasoning Architecture for Next-Generation Omnimodal Agents, 30%; No. RS-2020-II200882, (SW STAR LAB) Development of deployable learning intelligence via self-sustainable and trustworthy machine learning, 15%; No. RS-2022-II220480, Development of Training and Inference Methods for Goal Oriented Artificial Intelligence Agents, 15%; No. RS-2026-25522672, Development of Unified Reasoning Technology Mimicking Human Cognition for Hierarchical Understanding and Unbounded Problem Solving, 10%; and No. RS-2021-II211343, Artificial Intelligence Graduate School Program (Seoul National University), 10%], the Basic Science Research Program through the National Research Foundation of Korea (NRF) funded by the Ministry of Education (RS-2023-00274280, 10%), and the Advanced GPU Utilization Support Program funded by the Ministry of Science and ICT and supervised by NIPA (02-26-01-0285, 10%). Hyun Oh Song is the corresponding author.

## Impact Statement

This paper presents work whose goal is to advance the field of Machine Learning. There are many potential societal consequences of our work, none of which we feel must be specifically highlighted here.

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

# A. Agent Training and Implementation

## A.1. Training Loop

This section outlines the training procedure for the world model and the policy, which are trained concurrently through alternating update steps. The overall training loop is composed of the following steps:

1. **Environment interaction:** Execute the current policy in the real environment and store the resulting experiences in a replay buffer.

2. **Policy update on real data:** Update the policy using the most recent real environment experiences collected in Step 1. The policy is trained on the data over $E_{\text{env}}$ epochs, with each batch split into $B_{\text{policy}}$ minibatches due to memory constraints.

3. **Tokenizer training:** Sample experiences from the replay buffer to train the nearest neighbor tokenizer. The tokenizer is updated on $U_{\text{tokenizer}}$ batches of sample trajectories.

4. **World model training:** Sample experiences from the replay buffer to train the transformer world model. The world model is updated on $U_{\text{WM}}$ batches of sample trajectories, using $B_{\text{WM}}$ minibatches.

5. **Policy update in imagination:** For training steps $t > T_{\text{warmup}}$, generate $U_{\text{imag}}$ batches of imagined trajectories using the world model and the current policy, and update the policy on these synthetic rollouts. During the initial $T_{\text{warmup}}$ real environment interactions, this step is skipped to allow the world model to reach sufficient accuracy before generating imaginations. At the start of each imagination rollout, the policy uses $T_{\text{burn}}$ frames from the replay buffer to initialize its RNN hidden state.

The overall training loop is repeated until the agent has performed a total of $T_{\text{total}}$ real environment interactions.

## A.2. World Model Network

### A.2.1. WORLD MODEL ARCHITECTURE

Our transformer world model follows the GPT-2 architecture (Radford et al., 2019). The model operates over tokenized sequences that encode states and actions over $T$ consecutive frames. These tokens are first mapped to 128-dimensional embeddings via a learned embedding layer. Absolute positional embeddings are then added, followed by an initial dropout layer. The resulting embeddings are processed through a stack of three transformer blocks. Each block consists of the following components:

1. Layer normalization

2. Multi-head attention module, comprising:

   (a) Self-attention with a block causal mask. In the block causal mask, tokens within the same timestep are decoded in parallel (see Figure 6) (Dedieu et al., 2025).
   (b) A linear projection to the 128-dimensional embedding space
   (c) Dropout

3. Residual connection with the block input

4. Layer normalization

5. Feed-forward multilayer perceptron (MLP) composed of:

   (a) A hidden layer of dimension 512
   (b) GeLU activation
   (c) Dropout

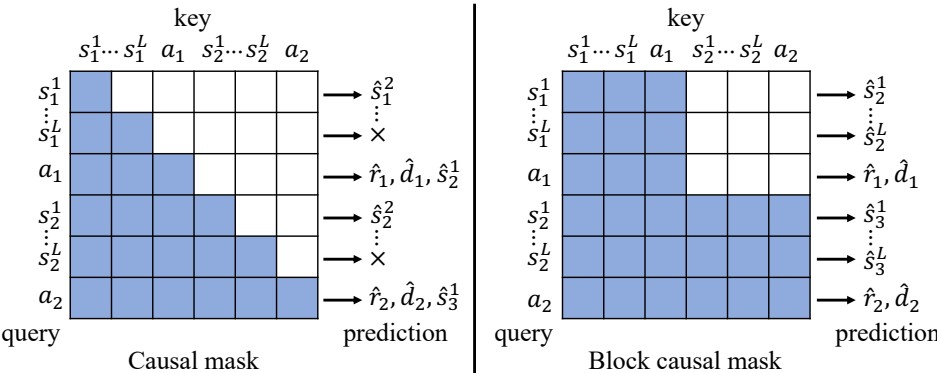

*Figure 6.* Comparison between the causal attention mask and the block causal attention mask. The token $s_t^i$ denotes the $i$-th state token at timestep $t$, $a_t$ denotes the action, $\hat{r}_t$ denotes the predicted reward, and $\hat{d}_t$ denotes the predicted done signal. Only two state tokens are shown per state for simplicity. (left) In the causal mask, each token attends to the tokens preceding it. The output embeddings of state token $s_t^i$ are used to predict the subsequent state token $s_t^{i+1}$. The reward $\hat{r}_t$ and done signal $\hat{d}_t$ are predicted from $a_t$, and the output of $s_t^L$ is unused. (right) In the block causal mask, all state and action tokens in the same timestep attend to each other, and they are used to predict the corresponding token in the next timestep ($a_t$ predicts $\hat{r}_t$ and $\hat{d}_t$). This allows each frame to be predicted in parallel rather than token-by-token.

After processing through the final block, the output undergoes a final layer normalization and is then passed to three separate prediction heads: one for the next state tokens, one for the reward signal, and one for the done signal. We denote the output embeddings as

$$(E_1^1, \ldots, E_1^{L+1}, E_2^1, \ldots, E_2^{L+1}, \ldots, E_T^1, \ldots, E_T^{L+1}).$$

where $L$ represents the number of state tokens per frame, and $E_t^i$ corresponds to the $i$-th output embedding at timestep $t$. These embeddings are routed to prediction heads as follows:

1. For $i \leq L$, the embedding $E_t^i$ is input to the observation head, an MLP comprising a 128-dimensional linear layer, a ReLU activation, and a final linear layer projecting to the codebook size $K$. The output logits define a categorical distribution over the $K$ possible values of the predicted state token $s_{t+1}^i$.

2. The embedding $E_t^{L+1}$, corresponding to the position of the action token, is passed to both the reward and done heads. Each head is an MLP consisting of a 128-dimensional linear layer, a ReLU activation, and a final linear layer projecting to two output classes. Although the Craftax-classic environment defines reward values of $-0.1$, $0.1$, and $1.0$, we follow Dedieu et al. (2025) and binarize the reward signal to improve stability, ignoring the $-0.1$ and $0.1$ cases.

The model is trained on trajectories of length $T_{\text{WM}}$ sampled from the replay buffer. The total loss is the sum of three components:

1. Cross-entropy loss over next-state token predictions (across $K$ classes).

2. Cross-entropy loss for binary reward classification (0 or 1).

3. Cross-entropy loss for done signal prediction.

Optimization is performed using the Adam optimizer with gradient norm clipping to stabilize training (Kingma & Ba, 2015). Hyperparameters for architecture and training are provided in Table 9.

### A.2.2. TRANSFORMER RoPE IMPLEMENTATION

3D RoPE is a standard positional encoding for spatio-temporal inputs. We adopt the formulation of VideoRoPE (Wei et al., 2025). While RoPE rotates pairs of embedding dimensions using frequencies based on a 1D position index, the 3D

*Table 9.* World model hyperparameters. Sweep range indicates the values tried per hyperparameter, with the final Value being chosen based on highest return.

| Area | Hyperparameter | Value | Sweep range |
|---|---|---|---|
| Architecture | Sequence length $T_{\mathrm{WM}}$ | 20 | |
| | State tokens per frame $L$ | 81 | |
| | Number of blocks | 3 | |
| | Number of attention heads | 8 | |
| | Embedding dimension | 128 | |
| | MLP hidden layer dimension | 512 | |
| | Dropout rate | 0.1 | |
| | Attention mask | Block causal | |
| | Inference with key-value caching | True | |
| Optimal transport | Distance cost coefficient $c_d$ | 0.6 | $\{0.0, 0.3, 0.6, 0.8\}$ |
| | Wildcard cost $c_w$ | 0.3 | $\{0.3, 0.6\}$ |
| | Sinkhorn regularization parameter $\epsilon$ | 0.00001 | $\{0.00001, 0.0001\}$ |
| | Sinkhorn iterations | 10 | $\{10, 100, 500\}$ |
| Training | Number of updates $U_{\mathrm{WM}}$ | 500 | |
| | Number of minibatches $B_{\mathrm{WM}}$ | 3 | |
| | Replay buffer size | 128,000 | |
| Optimization | Optimizer | Adam | |
| | Learning rate | 0.001 | |
| | Max norm for gradient clipping | 0.5 | |
| Tokenizer | Codebook size $K$ | 4096 | |
| | Single patch shape | $7 \times 7 \times 3$ | |
| | New code threshold $\tau$ | 0.75 | |
| | Number of updates $U_{\mathrm{tokenizer}}$ | 25 | |

version modulates the rotation amount based on three indices, two spatial and one temporal. We divide dimension pairs in a 3:1 ratio between spatial and temporal encoding. Pairs associated with lower rotation frequencies are used for temporal encoding and are rotated based on the temporal index. In contrast, pairs with higher rotation frequencies are used for spatial encoding. Given the 2D nature of spatial positions, spatial pairs are further split evenly between the horizontal and vertical axes. These are interleaved across the embedding dimension to ensure balanced representation. As a result, the axes contributing to rotation follow the pattern $(x, y, x, y, \ldots, x, y, t, t, \ldots, t)$, ordered by decreasing rotation frequency. The positional encoding is implemented by applying block-diagonal rotation matrices to the query and key vectors. The matrix $\mathbf{R}_{xy}$ applies higher-frequency rotations parameterized by the spatial coordinates $(x, y)$, while $\mathbf{R}_t$ applies lower-frequency rotations parameterized by the temporal index $t$. These rotations follow the standard RoPE formulation extended to two spatial dimensions and one temporal dimension.

$$
\boldsymbol{R}_{xy} = \begin{pmatrix}
\cos\theta_0 x & -\sin\theta_0 x & 0 & 0 & \cdots & 0 & 0 \\
\sin\theta_0 x & \cos\theta_0 x & 0 & 0 & \cdots & 0 & 0 \\
0 & 0 & \cos\theta_1 y & -\sin\theta_1 y & \cdots & 0 & 0 \\
0 & 0 & \sin\theta_1 y & \cos\theta_1 y & \cdots & 0 & 0 \\
\vdots & \vdots & \vdots & \vdots & \ddots & \vdots & \vdots \\
0 & 0 & 0 & 0 & \cdots & \cos\theta_{k-1} y & -\sin\theta_{k-1} y \\
0 & 0 & 0 & 0 & \cdots & \sin\theta_{k-1} y & \cos\theta_{k-1} y
\end{pmatrix}
$$

$$
\boldsymbol{R}_t = \begin{pmatrix}
\cos\theta_k t & -\sin\theta_k t & \cdots & 0 & 0 \\
\sin\theta_k t & \cos\theta_k t & \cdots & 0 & 0 \\
\vdots & \vdots & \ddots & \vdots & \vdots \\
0 & 0 & \cdots & \cos\theta_{D/2-1} t & -\sin\theta_{D/2-1} t \\
0 & 0 & \cdots & \sin\theta_{D/2-1} t & \cos\theta_{D/2-1} t
\end{pmatrix}
$$

Given a query vector $\mathbf{q}_i$ for token $i$ and a key vector $\mathbf{k}_j$ for token $j$, their rotary embeddings are obtained by applying the corresponding spatial and temporal rotations. Let $x(i)$, $y(i)$, and $t(i)$ denote the spatial and temporal coordinates of the $i$-th token. Then the transformed query and key vectors are:

$$\mathbf{q}_i' = \begin{pmatrix} \boldsymbol{R}_{x(i)y(i)} & \mathbf{0} \\ \mathbf{0} & \boldsymbol{R}_{t(i)} \end{pmatrix} \mathbf{q}_i$$

$$\mathbf{k}_j' = \begin{pmatrix} \boldsymbol{R}_{x(j)y(j)} & \mathbf{0} \\ \mathbf{0} & \boldsymbol{R}_{t(j)} \end{pmatrix} \mathbf{k}_j.$$

$$\mathbf{q}_i'^{\top} \mathbf{k}_j' = \mathbf{q}_i^{\top} \begin{pmatrix} \boldsymbol{R}_{x(j)-x(i),y(j)-y(i)} & \mathbf{0} \\ \mathbf{0} & \boldsymbol{R}_{t(j)-t(i)} \end{pmatrix} \mathbf{k}_j$$

As action tokens lack inherent spatial coordinates, assigning them fixed spatial positions would limit the effectiveness of 3D RoPE across the majority of embedding dimensions. To address this, spatial coordinates for action tokens are defined along the diagonal, $(t, t)$, where $t$ represents the temporal index. State tokens are assigned spatial coordinates offset from this diagonal, $(x + t, y + t)$, ensuring temporal alignment with action tokens while preserving spatial variation.

To avoid positional collisions between state and action tokens, they are given different temporal indices. That is, the state and action tokens $s_t^1, \ldots s_t^L, a_t, s_{t+1}^1, \ldots, s_{t+1}^L, a_{t+1}$ are given temporal indices $2t, \ldots, 2t, 2t + 1, 2(t + 1), \ldots 2(t + 1), 2(t + 1) + 1$. This staggered assignment ensures that each token occupies a unique spatio-temporal location, maintaining positional distinctiveness throughout the sequence.

### A.3. Policy Network

#### A.3.1. POLICY NETWORK ARCHITECTURE

We adopt the policy network architecture introduced in Dedieu et al. (2025), which comprises three primary components: a convolutional encoder, a recurrent neural network (RNN), and separate MLP heads for action and value prediction.

The convolutional encoder consists of three convolutional blocks with channel sizes [64, 64, 128]. Each block contains an instance normalization layer, a $3 \times 3$ convolutional layer with stride 1, a $3 \times 3$ max-pooling layer with stride 2, and two ResNet-style sub-blocks. Each ResNet block includes a ReLU activation, instance normalization, a 3×3 convolution with stride 1, and a skip connection to preserve the input. The encoder produces an output of shape $8 \times 8 \times 128$, which is flattened into a 8192-dimensional vector, denoted by $z$. The vector $z$ is then projected into a 256-dimensional representation through a ReLU activation, a linear layer, and layer normalization. This projected representation serves as input to a GRU recurrent module, which outputs a vector $y \in \mathbb{R}^{256}$ along with the updated hidden state $h \in \mathbb{R}^{256}$.

The action and value heads share an identical structure except for the final output projection. Each head takes the concatenated vector $[z, y]$ as input and applies a sequence of transformations: ReLU activation, layer normalization, a linear projection to 2048, another ReLU activation, and a residual block composed of two linear layers with ReLU activations. The output is passed through a final layer normalization, followed by the task-specific output projection—either to action logits or a scalar value estimate.

#### A.3.2. POLICY TRAINING

We follow the policy training procedure described in Dedieu et al. (2025), using Proximal Policy Optimization (PPO) (Schulman et al., 2017) as the underlying policy gradient algorithm.

Let the trajectory be denoted as $\tau = (o_{1:T+1}, a_{1:T}, r_{1:T}, d_{1:T}, h_{0:T})$, where $o_t$ represents the observations, $a_t$ the actions, $r_t$ the rewards, $d_t$ the done signals, and $h_t$ the hidden states of the RNN. At each timestep, PPO computes the value estimates $v_{1:T+1} = V_{\Phi_{\text{old}}}(o_{1:T+1})$ and the action probabilities $\pi_{\Phi_{\text{old}}}(a_t|o_t)$ under the current fixed parameters $\Phi_{\text{old}}$. The policy is optimized by minimizing the following PPO objective:

*Table 10.* Policy hyperparameters. Sweep range indicates the values tried per hyperparameter, with the final Value being chosen based on highest return.

| Area | Hyperparameter | Value | Sweep range |
|---|---|---|---|
| Environment | Environment interactions $T_{\text{total}}$ | 1,000,000 | |
| | Warmup interactions $T_{\text{warmup}}$ | 50,000 | {50k, 100k, 200k} |
| | Number of environments (batch size) | 48 | |
| | Rollout horizon in environment | 96 | |
| | Rollout horizon in imagination $T_{\text{WM}}$ | 20 | |
| | Burn-in horizon for RNN in imagination $T_{\text{burn}}$ | 5 | |
| Training | Number of updates in imagination $U_{\text{imag}}$ | 300 | {150, 300, 600, 1200} |
| | Number of epochs in environment $E_{\text{env}}$ | 4 | |
| | Number of epochs in imagination | 1 | |
| | Number of minibatches in environment $B_{\text{policy}}$ | 8 | |
| | Number of minibatches in imagination | 1 | |
| PPO | Discount factor $\gamma$ | 0.925 | |
| | TD weight $\lambda$ | 0.625 | |
| | Clipping value $\epsilon$ | 0.2 | |
| | TD loss coefficient $\lambda_{\text{TD}}$ | 2.0 | |
| | Entropy loss coefficient $\lambda_{\text{ent}}$ | 0.01 | |
| | PPO target discount factor $\alpha$ | 0.95 | |
| Optimization | Optimizer | Adam | |
| | Learning rate | 0.00045 | |
| | Max norm for gradient clipping | 0.5 | |

$$\mathcal{L}_{\text{PPO}}(\Phi) = \frac{1}{T} \sum_{t=1}^{T} \Big\{ - \min\left(p_t(\Phi)A_t, \text{clip}(p_t(\Phi))A_t\right)$$
$$+ \lambda_{\text{TD}}(V_\Phi(o_t) - q_t)^2$$
$$- \lambda_{\text{ent}}\mathcal{H}\left(\pi_\Phi(.|o_t)\right) \Big\}$$

where $p_t(\Phi)$ is the probability ratio $\frac{\pi_\Phi(a_t|o_t)}{\pi_{\Phi_{\text{old}}}(a_t|o_t)}$ and $\text{clip}(x)$ is the clipping function $\min(\max(x, 1-\epsilon), 1+\epsilon)$. Here, $A_t$ denotes a generalized advantage estimation, $q_t$ is a temporal difference (TD) target, and $\mathcal{H}$ is the entropy operator. The advantages $A_t$ and targets $q_t$ are computed as

$$A_t = \delta_t + (1 - \text{done}_t)\gamma\lambda A_{t+1},$$

$$q_t = A_t + v_t,$$

where $\delta_t = r_t + (1 - \text{done}_t)\gamma v_{t+1} - v_t$.

We incorporate two modifications to the standard PPO implementation:

- Generalized advantage estimates $A_t$ are standardized across training batches to stabilize learning.

- We track the moving average of the mean and standard deviation of $q_t$, with discount factor $\alpha$, and train the value function to predict the standardized targets.

Optimization is performed using the Adam optimizer with gradient norm clipping to stabilize training (Kingma & Ba, 2015). Hyperparameters for architecture and training are provided in Table 10.

---

**Algorithm 3** SINKHORN implementation

---

**Input:** Cost matrix $\boldsymbol{C}$,
Sinkhorn regularization parameter $\epsilon$,
Number of Sinkhorn iterations $T$
**Output:** Optimal transport plan $\boldsymbol{P}$

$\boldsymbol{K} = \exp(\boldsymbol{C}/\epsilon)$
Set uniform marginals: $\mathbf{r} = \frac{1}{\text{rows}(\boldsymbol{C})}$, $\mathbf{c} = \frac{1}{\text{cols}(\boldsymbol{C})}$
Initialize dual variables: $\mathbf{u} = \mathbf{1}$, $\mathbf{v} = \mathbf{1}$
**for** $t = 1$ **to** $T$ **do**
  $\mathbf{u} = \mathbf{r} \oslash (\boldsymbol{K}\mathbf{v})$ {$\oslash$ denotes element-wise division}
  $\mathbf{v} = \mathbf{c} \oslash (\boldsymbol{K}^\top\mathbf{u})$
**end for**
**Return** $\boldsymbol{P} = \text{diag}(\mathbf{u}) \cdot \boldsymbol{K} \cdot \text{diag}(\mathbf{v})$

---

## B. Optimal Transport Implementation

This section describes various implementation details of using optimal transport, including the definition of distance cost, using the outputs, and hyperparameter search. Algorithm 3 describes the Sinkhorn algorithm (Cuturi, 2013). We use the OTT-JAX library for our Sinkhorn solver implementation (Cuturi et al., 2022).

**Distance Cost**  For the affinity matrix $\boldsymbol{A}^{(prev)}$ in Equation 1, we use the same simple displacement cap as the distance penalty $D((x_i, y_i), (x_j, y_j))$ across all four benchmarks:

$$D((x_i, y_i), (x_j, y_j)) = \begin{cases} d, & \text{if } d \leq 4 \\ +\infty & \text{otherwise,} \end{cases}$$
$$\text{where } d = \|(x_i, y_i) - (x_j, y_j)\|_2^2.$$

The cap encodes a generic prior in visual RL: entities rarely move by more than a couple of token positions between consecutive frames. We use the same cap unchanged across Craftax-classic, Craftax, MinAtar, and Atari 100K, so $D$ does not need to be tuned to a specific environment's dynamics. The penalty can in principle be tailored to a particular environment when finer prior knowledge is available, but no such adaptation is needed for the results in this paper.

**Choosing Between Transformer and Optimal Transport Output**  Optimal transport provides an effective mechanism for reusing tokens from the previous frame. However, it is less effective in scenarios where novel tokens must be introduced, such as when the agent moves to a previously unexplored area. In such cases, optimal transport may fail to consistently route wildcard entries to the appropriate newly generated tokens. Conversely, the transformer world model is capable of freely generating new tokens as needed, but lacks a mechanism for directly reusing tokens from prior frames. Rather than committing to a single output modality, we adopt a hybrid strategy for Craftax-classic and Craftax that selects between optimal transport and transformer outputs based on spatial position. In Craftax-classic and Craftax, new visual content appears along the screen boundaries as the player explores previously unseen regions. Additionally, the inventory interface—fixed at the bottom of the screen—requires updates to token values without positional shifts. To accommodate these patterns, we apply the optimal transport output to the central region of the screen, where token reuse is most appropriate, while using the transformer's predictions for the screen edges and inventory regions, where new content is more likely. For MinAtar, which does not have special behavior at the edges, the optimal transport output is used directly for the entire screen.

## C. Craftax Hyperparameters

Following Dedieu et al. (2025), we change some hyperparameters for Craftax, as shown in Table 11. In particular, to accommodate the larger screen and additional tokens in memory, the batch size and replay buffer size are reduced.

*Table 11.* Hyperparameter differences between Craftax-classic and Craftax.

| Area | Hyperparameter | Craftax-classic | Craftax |
|------|----------------|-----------------|---------|
| Environment | Observation shape | $63 \times 63 \times 3$ | $130 \times 110 \times 3$ |
| | Number of possible actions | 17 | 43 |
| | Number of possible achievements | 22 | 226 |
| | Number of environments (batch size) | 48 | 16 |
| Tokenizer | Single patch shape | $7 \times 7 \times 3$ | $10 \times 10 \times 3$ |
| Architecture | State tokens per frame $L$ | 81 | 143 |
| Training | Replay buffer size | $128,000$ | $48,000$ |

## D. MinAtar Return Curves and Hyperparameters

Figure 7 shows the return curves of ITC and baseline Dedieu et al. (2025) for each game in MinAtar. ITC outperforms the baseline in every game. Table 12 lists the hyperparameters for MinAtar with different values compared to Craftax-classic. All hyperparameter changes follow Dedieu et al. (2025), except the hyperparameters specific to optimal transport. Also following Dedieu et al. (2025), the policy encoder uses layer normalization and the Swish activation function, and actor and value networks share weights except in their final linear layers (Ba et al., 2016; Ramachandran et al., 2017).

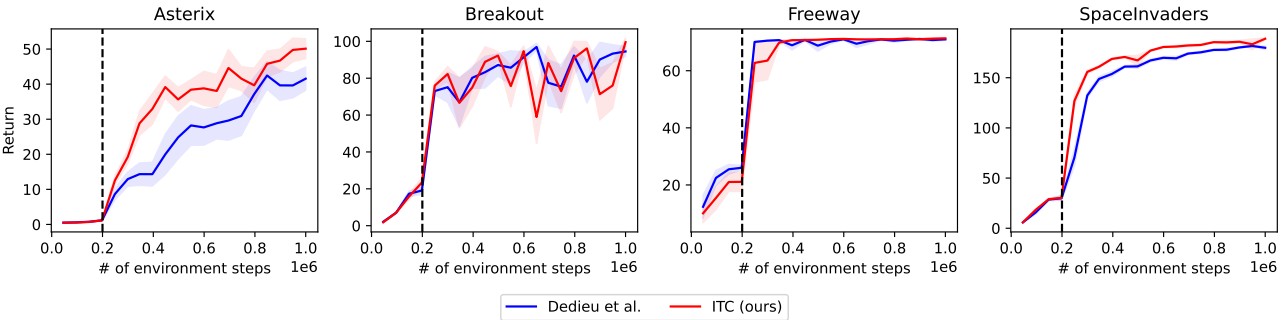

*Figure 7.* Return curves for MinAtar. Shading indicates standard error among multiple seeds. The vertical dashed lines indicate the start of training in imagination after $T_{\text{warmup}}$ interactions.

## E. Additional Atari 100K Results and Hyperparameters

Table 13 shows the average return for each of 26 games in Atari 100K. We follow the Simulus hyperparameter settings (Cohen et al., 2025), and additionally include our proposed hyperparameters: distance cost coefficient $c_d$ and wildcard cost $c_w$, set to $0.05$ and $0.01$, respectively.

*Table 12.* Hyperparameter differences between Craftax-classic and MinAtar. $K$ is the number of object types for each game (4 for Asterix, 4 for Breakout, 7 for Freeway, and 6 for SpaceInvaders). The number of actions $A$ is 5 for Asterix, 3 for Breakout, 3 for Freeway, and 4 for SpaceInvaders.

| Area | Hyperparameter | Craftax-classic | MinAtar |
|---|---|---|---|
| Environment | Observation shape | $63 \times 63 \times 3$ | $10 \times 10 \times K$ |
| | Number of possible actions | 17 | $A$ |
| | Warmup interactions $T_{\text{warmup}}$ | 50,000 | 200,000 |
| Tokenizer | Single patch shape | $7 \times 7 \times 3$ | $2 \times 2 \times K$ |
| Architecture | State tokens per frame $L$ | 81 | 25 |
| Optimal transport | Distance cost coefficient $c_d$ | 0.6 | 0.2 |
| | Wildcard cost $c_w$ | 0.3 | 0.05 |
| Training | Number of world model updates $U_{\text{WM}}$ | 500 | 2000 |
| | Number of policy updates in imagination $U_{\text{imag}}$ | 300 | 2000 |
| | Coefficient for reward prediction loss | 1 | 10 |
| | Coefficient for done prediction loss | 1 | 10 |
| PPO | Discount factor $\gamma$ | 0.925 | 0.95 |
| | TD weight $\lambda$ | 0.625 | 0.75 |
| | Entropy loss coefficient $\lambda_{\text{ent}}$ in imagination | 0.01 | 0.05 |
| | PPO target discount factor $\alpha$ | 0.95 | 0.925 |

*Table 13.* Mean returns on the 26 games of the Atari 100k benchmark followed by averaged human-normalized performance metrics. Each game score is computed as the average of 5 runs with different seeds. Bold face mark the best score.

| Game | Random | Human | DreamerV3 | STORM | DIAMOND | Simulus | ITC (ours) |
|---|---|---|---|---|---|---|---|
| Alien | 227.8 | 7127.7 | 959.4 | **983.6** | 744.1 | 687.2 | 727.4 |
| Amidar | 5.8 | 1719.5 | 139.1 | 204.8 | **225.8** | 102.4 | 144.7 |
| Assault | 222.4 | 742.0 | 705.6 | 801.0 | 1526.4 | **1822.8** | 1455.2 |
| Asterix | 210.0 | 8503.3 | 932.5 | 1028.0 | **3698.5** | 1369.1 | 1610.2 |
| BankHeist | 14.2 | 753.1 | **648.7** | 641.2 | 19.7 | 347.1 | 370.6 |
| BattleZone | 2360.0 | 37187.5 | 12250.0 | 13540.0 | 4702.0 | 13262 | **13590.4** |
| Boxing | 0.1 | 12.1 | 78.0 | 79.7 | 86.9 | **93.5** | 91.9 |
| Breakout | 1.7 | 30.5 | 31.1 | 15.9 | 132.5 | **148.9** | 75.4 |
| ChopperCommand | 811.0 | 7387.8 | 410.0 | 1888.0 | 1369.8 | 3611.6 | **4720.8** |
| CrazyClimber | 10780.5 | 35829.4 | 97190.0 | 66776.0 | **99167.8** | 93433.2 | 95114.2 |
| DemonAttack | 152.1 | 1971.0 | 303.3 | 164.6 | 288.1 | 4787.6 | **4814.6** |
| Freeway | 0.0 | 29.6 | 0.0 | 0.0 | **33.3** | 31.9 | 32.0 |
| Frostbite | 65.2 | 4334.7 | 909.4 | **1316.0** | 274.1 | 358.4 | 266.5 |
| Gopher | 257.6 | 2412.5 | 3730.0 | **8239.6** | 5897.9 | 4363.2 | 4967.8 |
| Hero | 1027.0 | 30826.4 | **11160.5** | 11044.3 | 5621.8 | 7466.8 | 6603.0 |
| Jamesbond | 29.0 | 302.8 | 444.6 | 509.0 | 427.4 | 678.0 | **1126.9** |
| Kangaroo | 52.0 | 3035.0 | 4098.3 | 4208.0 | 5382.2 | 6656.0 | **9725.4** |
| Krull | 1598.0 | 2665.5 | 7781.5 | 8412.6 | **8610.1** | 6677.3 | 6446.7 |
| KungFuMaster | 258.5 | 22736.3 | 21420.0 | 26182.0 | 18713.6 | **31705.4** | 19948.8 |
| MsPacman | 307.3 | 6951.6 | 1326.9 | **2673.5** | 1958.2 | 1282.7 | 1133.7 |
| Pong | -20.7 | 14.6 | 18.4 | 11.3 | **20.4** | 19.9 | 18.5 |
| PrivateEye | 24.9 | 69571.3 | 881.6 | **7781.0** | 114.3 | 100.0 | 726.3 |
| Qbert | 163.9 | 13455.0 | 3405.1 | **4522.5** | 4499.3 | 2425.6 | 3194.6 |
| RoadRunner | 11.5 | 7845.0 | 15565.0 | 17564.0 | 20673.2 | 24471.8 | **23535.4** |
| Seaquest | 68.4 | 42054.7 | 618.0 | 525.2 | 551.2 | **1800.4** | 1273.8 |
| UpNDown | 533.4 | 11693.2 | 7567.1 | 7985.0 | 3856.3 | 10416.5 | **12901.1** |
| #Superhuman (↑) | 0 | N/A | 9 | 9 | 11 | **13** | **13** |
| Mean (↑) | 0.000 | 1.000 | 1.124 | 1.222 | 1.459 | **1.645** | 1.616 |
| Median (↑) | 0.000 | 1.000 | 0.485 | 0.425 | 0.373 | **0.982** | 0.978 |
| IQM (↑) | 0.000 | 1.000 | 0.487 | 0.561 | 0.641 | 0.990 | **1.092** |
| Optimality Gap (↓) | 1.000 | 0.000 | 0.510 | 0.472 | 0.480 | 0.412 | **0.376** |

