# OpenReview forum: "Identifiable Token Correspondence for World Models"
_ICML.cc/2026/Conference — ICML 2026 regular_

### Official Review · Reviewer_JWqb · 2026-03-11

**Soundness:** 2
**Presentation:** 2
**Significance:** 2
**Originality:** 2
**Overall Recommendation:** 3
**Confidence:** 4

**Summary:**

This pager aims to address the time consistency challenge with world models. The key idea is to model explicitly the correspondence of tokens between adjacent frames so that the sharing of content across frames can better captured, in comparison to the default design that leave this correspondence to be modelled *implicitly* (or pure data driven). The method is to append an extra layer using the optimal transport based token assignment process, where two options are designed: copy a token from the previous frame if shared, otherwise generating a new token for predicting the next frame. Evaluation has been done on some game based benchmarks to validate the efficacy of this proposed method.

**Compliance With Llm Reviewing Policy:**

Affirmed.

**Final Justification:**

I thank the responses made by authors, which indeed addressed many of my initial concerns.

At this moment, I am still confused why and not fully get that the choice of the wildcard-token penalty can be just fixed but not content dependent, given the nature of this proposed method -- copy the same tokens if no change from the current frame to the next. This is really the major concern as it seems to be contradictory.

Given this issue, I would not champion the acceptance of this work and am leaned to have the authors offer more solid work next.

**Key Questions For Authors:**

The authors should specify these key issues:

- what is the baseline method used for ITC, including training setting etc.

- Is the current comparison fair and in what sense?

- Can explicit modelling of token correspondence across frames bring positive result and how much, in what setting?

- Can this proposed idea be integrated into different world models?

**Limitations:**

Yes

**Strengths And Weaknesses:**

**Strengths**

- The idea is intuitive that explicitly modelling the correspondence of tokens between frames may help with token prediction in world model training. This could be likely more effective than the previous design of leaving this to be modelled implicitly (the baseline or competitor).

**Weaknesses**

- *Writing issues*: In Abstract, the authors do not well explained the connection between temporal inconsistency and token correspondence. A more detailed discussion should be made to capture the overall idea of this work.

- *Position issue*: It is unclear how this proposed method is positioned among the literature. From my understanding, this method serves as a general training recipe that can be integrated with existing world model during their training time. However, this is not made clearly enough throughout the whole paper.

- Technically, I do not see any new designs with this method, but just taking existing algorithms to implement the proposed idea. I would say this is still acceptable but at least not able to claim strong novelty in terms of modelling.

- In Table 1, it is clear that what are the competitors such as DART, DreamerV3, IRIS etc. But more importantly, what is the baseline method for ITC, the proposed method? Please note the difference of competitor and baseline -- which are mixed here in the writing and thus confusing. To be concrete, baseline is the method the authors build their correspondence search component on top, while competitors are other methods that address the same problem but would be different in many aspects including the backbone network, optimisation loss function, etc.

- Following the above, the current results (Table 1, Figure 3) just show the comparison at the system level, but not able to draw any conclusion on the research question - weather explicitly modelling the token correspondence across frames helps in comparison to the implicit modelling.

---

> ### Author Rebuttal · Authors · 2026-03-31
>
> We sincerely thank you for reviewing our paper and for your constructive feedback. We agree that the presentation can be clearer, and we will revise the writing accordingly. We also appreciate your recognition that the method is intuitive and that the paper provides a clear theoretical motivation for its improvements over the baselines.
>
> ---
>
>
> > **[Weakness 1]** Writing issues: In Abstract, the authors do not well explained the connection between temporal inconsistency and token correspondence. ...
>
> The key idea is that temporal inconsistency arises when persistent entities are regenerated independently at each step without explicitly deciding whether they should persist from the previous frame. This can lead to duplication, disappearance, or identity drift over rollouts. ITC reduces this problem by explicitly modeling which next-frame tokens should be copied and which should be newly generated. We will strengthen this explanation in the revised paper and connect it more directly to the qualitative visualizations.
>
> ---
>
>
> > **[Weakness 2]** Position issue: ... From my understanding, this method serves as a general training recipe that can be integrated with existing world model during their training time. However, this is not made clearly enough throughout the whole paper.
>
> ITC is not meant to be a standalone world model or a separate training recipe; rather, it is a decoding method that can be attached to an existing token-based world model. It does not require redefining the training objective and introduces only negligible runtime overhead in our measured setting. In the revision, we will make this positioning explicit throughout the paper.
>
> ---
>
>
> > **[Weakness 3]** Technically, I do not see any new designs with this method, but just taking existing algorithms to implement the proposed idea. ...
>
> The novelty is in using OT as a decoding mechanism for next-frame prediction in token-based world models, with a transport graph that explicitly represents copy-versus-generate choices, a wildcard mechanism for new content, distance-aware affinity design, and a binarization step to obtain discrete next-state assignments. In other words, the contribution is a new decoding formulation for world modeling. We’d like to note that other reviewers (K2Ga and XqWA) agree with us that our method is novel.
>
> ---
>
>
> > **[Weakness 4]** In Table 1, it is clear that what are the competitors such as DART, DreamerV3, IRIS etc. But more importantly, what is the baseline method for ITC, the proposed method? Please note the difference of competitor and baseline ...
>
> > **[Question 1]** what is the baseline method used for ITC, including training setting etc.
>
> > **[Question 2]** Is the current comparison fair and in what sense?
>
> On Craftax-classic, Craftax, and MinAtar, the direct baseline for ITC is *Dedieu et al. (2025)*. On Atari 100K, the direct baseline is *Simulus*, because that is the relevant token-based state-of-the-art world model in that benchmark. We reproduced the underlying baseline system to match reported performance and then added the OT decoding component on top of the same codebase. This makes the performance comparison of ITC vs. the underlying baseline model a fair comparison. By contrast, methods such as DreamerV3, IRIS, Delta-IRIS, and DART are competitors addressing the broader problem with different backbones and training designs. We will revise Table 1 and the surrounding text to separate these notions explicitly.
>
> ---
>
>
> > **[Weakness 5]** Following the above, the current results (Table 1, Figure 3) just show the comparison at the system level, but not able to draw any conclusion on the research question - weather explicitly modelling the token correspondence across frames helps in comparison to the implicit modelling.
>
> > **[Question 3]** Can explicit modelling of token correspondence across frames bring positive result and how much, in what setting?
>
> Under matching settings, ITC improves both token prediction accuracy (**Table 2**) and downstream policy performance (**Table 1**) relative to the reproduced baseline. This shows that explicit modeling of token correspondence is beneficial beyond implicit token generation alone. It leads to better imagination quality over rollouts, which provides informative experience to policy learning. We will strengthen this relation in the revision.
>
> ---
>
>
> > **[Question 4]** Can this proposed idea be integrated into different world models?
>
> Yes, for token-based world models. We already demonstrate this in two distinct instantiations: one built on the *Dedieu et al. (2025)* transformer-style world model for Craftax-classic, Craftax, and MinAtar, and another built on *Simulus* SSM-based world model for Atari 100K. This shows that ITC is not tied to a single backbone implementation. In the revision, we will clarify that the method is intended as a generally attachable decoding module for token-based world models rather than a one-off standalone architecture.

---

> > ### Author Rebuttal · Reviewer_JWqb · 2026-04-03
> >
> > Thank the authors for the responses, which have clarified most of the issues raised. Clearly this paper needs to be revised significantly in all sections, which is double but not trivial.
> >
> > Concurring with other reviewers, the current evaluation needs to be improved with more diverse data/tasks. This is also indicated by the ablation of the balance between copying a token from the previous frame and generating a new one. Intuitively this parameter is content dependent but not come as a single optimal value -- the current test result cannot explain this nature, even potentially misleading.

---

> > > ### Author Response · Authors · 2026-04-08
> > >
> > > In regards to choosing the wildcard penalty hyperparameter $c_w$, the performance may improve by manually tuning $c_w$ to different content (or individual game, in the case of Atari 100K), but such manual hyperparameter tuning would make our method harder to use, more brittle, and less generalizable. In practice, choosing a single fixed value hyperparameter is sufficient for superior performance over baselines, which is simpler and more robust than tuning the hyperparameter to different content (or different games within Atari 100K). In addition, our method is robust to different choices of $c_w$, as shown by the small performance difference between values in Table 10 of Appendix B.
> > >
> > > We note that reviewers K2Ga, XqWA, and JWqb raised the same concern about environment diversity; we provide a unified response.
> > >
> > > We agree that adding more environments will make our paper stronger. However, to make a fair comparison with the baselines, we cover all possible visual environments that existing baselines did and ITC shows consistent improvement across various environments compared to the baselines. Note that our paper includes more diverse environments than previous model-based reinforcement learning papers in this area. For example, Dedieu et al. was initially published with only Craftax and MinAtar environments (not Atari 100K). IRIS and DART only evaluate on the Atari 100K environment, and delta-IRIS only evaluates on Crafter and Atari 100K.
> > >
> > > We also want to note that our method and evaluation are not limited to grid-like environments. Using a VQ-VAE tokenizer as we did in Atari 100K, continuous pixel observations turn into latent tokens with spatial relationships, and our method is modelling such latent tokens’ correspondences, which supports all non-grid-like environments.
> > >
> > > We want to share the visualization of some games in Atari 100k, to illustrate that these environments are visually diverse:
> > > - BattleZone (3D-like dynamics as in DOOM): https://www.youtube.com/watch?v=eFqk9tyJ5i4
> > > - PrivateEye (frequent scene changes and sprites’ teleport): https://youtu.be/55S4PJpJ59c?si=o324ZTOo2iX3zZJZ
> > > - ChopperCommand (dynamic movements and camera walks): https://www.youtube.com/watch?v=YBTzXSQQNJU
> > >
> > > In conclusion, we agree that additional environments can be supplementary evidence to prove superior performance of our method, but we believe we already covered possible fair comparisons to baselines and proved superior performance on visually diverse environments.

---

### Official Review · Reviewer_XqWA · 2026-03-11

**Soundness:** 2
**Presentation:** 3
**Significance:** 2
**Originality:** 3
**Overall Recommendation:** 3
**Confidence:** 3

**Summary:**

This paper investigates a novel architecture for world models. It proposes an autoregressive transformer approach in which the input is transformed into a set of tokens, and a transformer is applied to predict the representations of the next frames from past tokens in a block-causal fashion. The paper’s main contribution is the application of an optimal-transport layer to leverage the strong temporal correlation present in video. This layer uses optimal-transport optimization (with Sinkhorn) to copy tokens from the next frames, or to leverage newly generated tokens that are associated with a “wildcard” token penalty.


The authors validate their approach on various RL game environments such as Craftax, MinAtar, and Atari-100k, applying their method on top of two different world-model baselines. The results show that (1) their ITC contribution consistently outperforms the baselines in return and score, and (2) ITC does not add significant compute overhead compared with the baseline.

**Compliance With Llm Reviewing Policy:**

Affirmed.

**Final Justification:**

As highlighted in my review, while the paper propose an interesting idea, the experimental protocol focuses on environment with limited visual diversity (Atari). It's unclear how the method would generalize to more complex environment.

**Key Questions For Authors:**

## Questions for the authors (from the weaknesses)

1. **Scaling / architecture size**
   - What is the scale of the current model (e.g., parameter count, token count per frame, context length, training compute), and do you expect the proposed optimial transport block to scale favorably as model size increases?
   - How does the method’s computational and memory cost scale with the number of tokens used to represent each frame?

2. **Generality beyond sprite-based environments**
   - Since the experiments focus on sprite-based environments, how do you expect the approach to perform on visually and dynamically richer continuous-control settings (e.g., DeepMind Control, RoboCasa)? Do you have preliminary results or plans to evaluate there?

3. **Role of the wildcard-token penalty \(c_w\)**
   - How sensitive are results to the choice of the wildcard-token penalty \(c_w\)? Can you share an ablation (or guidance) showing how performance changes as \(c_w\) varies and how you select it in practice?

**Limitations:**

No limitation section is present. It would be nice to add one highlighting potential weaknesses of the approach.

**Strengths And Weaknesses:**

Strengths:
- The paper’s contribution is clear and well motivated.
- The authors explore various RL environments (Craftax, MinAtar, Atari-100k) and show that their approach provides consistent performance improvements.
- The authors demonstrate that their contribution adds only limited computational overhead on top of the baselines.

Weaknesses:
- It is unclear from the paper what the scale of the current architecture is, and whether the proposed contribution (i.e., the proposed architectural block) would scale with model size. Additionally, how does the computational cost evolve with the number of tokens used to represent a frame?
- All environments considered in the paper use sprite-based visuals, which limits both visual complexity and dynamics. It would be good to demonstrate that the approach generalizes to more continuous environments (e.g., DeepMind Control, RoboCasa).
-  c_w, he wildcard-token penalty in the affinity matrix, seems to be an important hyperparameter because it controls the trade-off between copying previous tokens and generating new ones. It would be useful to include an ablation study of its effect in the main paper.

---

> ### Author Rebuttal · Authors · 2026-03-31
>
> Thank you so much for your review and your well-thought-out feedback. We appreciate that you considered our contribution well-motivated, our performance improvements consistent across diverse environments, and our method computationally efficient.
>
> ---
>
>
> > **[Weakness 1]** It is unclear from the paper what the scale of the current architecture is, and whether the proposed contribution (i.e., the proposed architectural block) would scale with model size. Additionally, how does the computational cost evolve with the number of tokens used to represent a frame?
>
> > **[Question 1]** What is the scale of the current model (e.g., parameter count, token count per frame, context length, training compute), and do you expect the proposed optimial transport block to scale favorably as model size increases? How does the method’s computational and memory cost scale with the number of tokens used to represent each frame?
>
> The model consists of 602K parameters for the tokenizer, 1.70M for the world model, and 54.5M for the policy network, identical to the *Dedieu et al. (2025)* baseline. Note that ITC does not introduce any additional model parameters and does not rely on the model size. The OT computation time for ITC scales quadratically with the number of tokens per frame as discussed in **Section 2.4**, but not with the context length or model size. In practice, the overhead in our current setting is minimal, as reported in **Section 4.1 - Compute Time Analysis**. ITC increases policy training time by only 2.8%. We will update our paper to clarify the scale of the model.
>
>
> ---
> > **[Weakness 2]** All environments considered in the paper use sprite-based visuals, which limits both visual complexity and dynamics. It would be good to demonstrate that the approach generalizes to more continuous environments (e.g., DeepMind Control, RoboCasa).
>
> > **[Question 2]** Since the experiments focus on sprite-based environments, how do you expect the approach to perform on visually and dynamically richer continuous-control settings (e.g., DeepMind Control, RoboCasa)? Do you have preliminary results or plans to evaluate there?
>
> We include Atari 100K benchmark to evaluate ITC on diverse visual complexity and dynamics. Atari 100K is not limited to sprite-based visuals, but includes raw-pixel games with room transitions, perspective changes, and more diverse visual structure, such as BankHeist, BattleZone, and PrivateEye. ITC also achieves superior performance on those games, as shown in **Table 13** in Appendix E.
>
> The main requirement for ITC is not “sprites” but persistent visual structure across adjacent frames that can either be reused or regenerated, which is a common characteristic of visual environments. We have not yet evaluated DeepMind Control or RoboCasa in this submission, and we will state this clearly as future work.
>
> \
> *Part of Table 13: Mean returns on the 26 games of the Atari 100k benchmark.*
> | Game | Simulus | ITC (ours) |
> |-|-|-|
> | BankHeist | 347.1 | 370.6 |
> | BattleZone | 13262 | 13590.4 |
> | PrivateEye | 100.0 | 726.3 |
>
> ---
>
>
> > **[Weakness 3]** $c_w$, the wildcard-token penalty in the affinity matrix, seems to be an important hyperparameter because it controls the trade-off between copying previous tokens and generating new ones. It would be useful to include an ablation study of its effect in the main paper.
>
> > **[Question 3]** How sensitive are results to the choice of the wildcard-token penalty ($c_w$)? Can you share an ablation (or guidance) showing how performance changes as ($c_w$) varies and how you select it in practice?
>
> The relevant ablation is already provided in Appendix B, where **Table 10** shows that ITC is robust across a reasonable range of $c_w$ values. We will surface this more clearly in the rebuttal and the revised paper.
>
>
> Conceptually, $c_w$ controls the balance between copying a token from the previous frame and generating a new one. The results show that the method is robust to this choice. We will also clarify that setting the penalty to zero effectively recovers the baseline decoding behavior, which explains why the baseline comparison is the most meaningful reference point.
>
> \
> *Table 10: Average returns and scores with respect to $c_w$, a constant penalty for using a wildcard token.*
>
> | $c_w$ | Return (\%) | Score (\%) |
> |-------|-------------|------------|
> | 0.3   | 72.46       | 35.60      |
> | 0.6   | 71.60       | 35.41      |

---

> > ### Author Rebuttal · Reviewer_XqWA · 2026-04-01
> >
> > Thank you for rebuttal. While the paper propose an interesting idea, the experimental protocol is limited experimental setup, focusing on environment with limited visual diversity. It's unclear how the method would generalize to more complex environment which are less grid-like.

---

> > > ### Author Response · Authors · 2026-04-08
> > >
> > > We note that reviewers K2Ga, XqWA, and JWqb raised the same concern about environment diversity; we provide a unified response.
> > >
> > > We agree that adding more environments will make our paper stronger. However, to make a fair comparison with the baselines, we cover all possible visual environments that existing baselines did and ITC shows consistent improvement across various environments compared to the baselines. Note that our paper includes more diverse environments than previous model-based reinforcement learning papers in this area. For example, Dedieu et al. was initially published with only Craftax and MinAtar environments (not Atari 100K). IRIS and DART only evaluate on the Atari 100K environment, and delta-IRIS only evaluates on Crafter and Atari 100K.
> > >
> > > We also want to note that our method and evaluation are not limited to grid-like environments. Using a VQ-VAE tokenizer as we did in Atari 100K, continuous pixel observations turn into latent tokens with spatial relationships, and our method is modelling such latent tokens’ correspondences, which supports all non-grid-like environments.
> > >
> > > We want to share the visualization of some games in Atari 100k, to illustrate that these environments are visually diverse:
> > > - BattleZone (3D-like dynamics as in DOOM): https://www.youtube.com/watch?v=eFqk9tyJ5i4
> > > - PrivateEye (frequent scene changes and sprites’ teleport): https://youtu.be/55S4PJpJ59c?si=o324ZTOo2iX3zZJZ
> > > - ChopperCommand (dynamic movements and camera walks): https://www.youtube.com/watch?v=YBTzXSQQNJU
> > >
> > > In conclusion, we agree that additional environments can be supplementary evidence to prove superior performance of our method, but we believe we already covered possible fair comparisons to baselines and proved superior performance on visually diverse environments.

---

### Official Review · Reviewer_K2Ga · 2026-03-12

**Soundness:** 3
**Presentation:** 3
**Significance:** 2
**Originality:** 3
**Overall Recommendation:** 3
**Confidence:** 4

**Summary:**

This paper aims to resolve the temporal inconsistency problem of transformer-based world models. The authors proposed Identifiable Token Correspondence (ITC) method where each token in the next frame prediction either comes from the transformer predictions or a copy of a token from the previous frame. This problem is solved using optimal transport on the cross-frame token correspondence. Experiment results on multiple pixel-based video games dataset reached good performance improvement over the previous models.

**Compliance With Llm Reviewing Policy:**

Affirmed.

**Final Justification:**

Thanks the authors for the rebuttal. The justification indicate my main concern about the paper, that is the lack of solid prove of the proposed method can work across more diverse set of environment, is valid. This issue is also raised by all the other reviewers. Therefore, I remain my previous score.

**Key Questions For Authors:**

1. How should Table 2 prediction accuracy be interpreted relative to downstream return / score? If this metric is meant to support the main claim, please clarify why it is informative and whether it is robust to tokenizer design choices.

2. How dependent is ITC on the simplicity and stability of the current token space? In particular, I wish the authors could explain how this method could work in VAE-based latent video model settings, where latent tokens are typically more compressed and entangled.

3. Can the authors better disentangle the roles of OT, binarization, 3D RoPE, and hybrid decoding in the final performance gains?

**Limitations:**

The paper should more clearly discuss that the current evidence is limited to highly discrete, tokenized 2D environments, and does not yet establish applicability to more realistic visual world models.

**Strengths And Weaknesses:**

## Soundness:

**Strengths**

The main motivation of this paper is reasonable: standard transformer-based next-token prediction for world models does not constraint on token identify across time, leading to a lot of hallucination, lose of information or jittering issues. The proposed method used optimal transport for world-model decoding to explicitly resolve this issue. Experiment results showed the performance has been improved.

**Weakness**

I have some concerns about how strongly the result supports the claim in more general cases:

1. The accuracy protocol in Table 2 is not very convincing, which is based on the exact one-step next-state matching. But there the paper does not clearly indicate how is this state matching relates to the downstream return/score; nor how sensitive it is to tokenizer patch size or the coodbook design.

2. The result is still limited to highly discrete, token-friendly 2D grid environments: the method relies on a discrete tokenizer, local token displacement assumptions. Therefore, the paper shows effectiveness in structured tokenized settings, but it does not yet establish that the same mechanism would remain valid in more realistic photorealistic VAE-based latent video model settings, like WAN-based models.

## Presentation:

The paper is generally well written and easy to follow. The high-level intuition is clear, and the qualitative examples help explain what type of temporal inconsistency the method is trying to fix. The overall structure is standard and readable.

**Weakness**

1. The framing is somewhat stronger than what the paper actually proves. In particular, the title term “Identifiable” sounds stronger than the evidence provided in the paper. Furthermore, as there is no experiment on VAE-based or photorealistic RGB tokenizer models,


## Significant:

**Strengths**

The paper studies an important problem for world models. Temporal consistency is a real bottleneck for long-horizon rollout and imagination-based training, so improving token identity preservation across time is a meaningful direction. I believe the proposed idea of explicit token correspondence could inspire follow-up work in structured decoding for world models.

**Weakness**

1. As mentioned, the demonstrated significance is currently limited by the experimental scope. All results are in discrete and highly structured environments.

2. Because the paper does not yet provide evidence in more realistic visual settings, it is hard to judge how much impact the method will have beyond the current class of tokenized low-pixel game-like environments. Even for old 3D-like games (e.g., DOOM), I doubt how much this idea can help the world models like GameNGen.


## Originality:

**Strengths**


The main originality of the paper is to use optimal transport as an explicit token correspondence mechanism inside world-model decoding. This is a meaningful conceptual shift: next-frame prediction is treated not only as token generation, but also as token identity assignment across time. I think this is a real and non-trivial idea.

**Weakness**

Some supporting elements, such as 3D RoPE, do not seem especially original and should not be counted as major novelty.

---

> ### Author Rebuttal · Authors · 2026-03-31
>
> Thank you very much for your detailed review. We appreciate that you understood our method to be well-motivated and our experimental results to be supportive of the improvements brought by our method.
>
> ---
> >**[Soundness 1]** ... Table 2 is not very convincing, which is based on the exact one-step next-state matching. But there the paper does not clearly indicate how is this state matching relates to the downstream return/score; nor how sensitive it is to tokenizer patch size or the coodbook design.
>
> >**[Question 1]** How should Table 2 prediction accuracy be interpreted relative to downstream return / score? ...
>
> Our intent in Table 2 was to use exact next-state matching as a diagnostic of world-model fidelity, not as a standalone claim. Better one-step prediction leads to better imagined rollouts, which in turn improves policy learning. This is also what we observe empirically: ITC improves token-level prediction accuracy and also improves downstream return and score. We will revise the text around Table 2 to make this relationship explicit.
>
> Our results are not tied to a single tokenizer configuration. We evaluate with different patch sizes and tokenizer regimes across benchmarks, and the gains remain consistent. For example, Craftax-classic, Craftax-full, and MinAtar use different patch granularities, and Atari 100K is evaluated with a VQ-VAE tokenizer built on Simulus. We will make this robustness point explicit in the revision.
>
> ---
> >**[Soundness 2]** The result is still limited to highly discrete, token-friendly 2D grid environments: the method relies on a discrete tokenizer ...
>
> >**[Significant 1]** ... All results are in discrete and highly structured environments.
>
> >**[Significant 2]** ... Even for old 3D-like games (e.g., DOOM), I doubt how much this idea can help the world models like GameNGen.
>
> >**[Question 2]** ... I wish the authors could explain how this method could work in VAE-based latent video model settings, where latent tokens are typically more compressed and entangled.
>
> >**[Limitations]** The paper should more clearly discuss that the current evidence is limited to highly discrete, tokenized 2D environments ...
>
> We acknowledge that our paper focuses on token-based world models rather than all world-model settings (like GameNGen, which is diffusion-based). ITC is designed for discrete token spaces where token reuse across adjacent frames is meaningful. We will state this limitation more clearly in the paper.
>
> At the same time, the current empirical scope is broader than “simple 2D grid environments.” Atari 100K uses raw-pixel observations, and includes games like BattleZone, PrivateEye, and ChopperCommand that exhibit 3D-like perspective effects (similar to DOOM), abrupt visual changes, and continuous diagonal movement. ITC shows better performance in such environments: ITC outperforms its baseline Simulus in all 3 mentioned games, and additionally outperforms all previous baselines in BattleZone and ChopperCommand (see **Table 13** in Appendix E).
>
> More broadly, discrete tokenization remains a strong design choice and is used in most prior work in model-based RL (IRIS, delta-IRIS, DART, Dedieu et al. (2025), Simulus, etc.), which is why we focus on that regime here. Also, note that we used the VQ-VAE tokenizer to match the same setting with the baseline, Simulus, for Atari 100k. The tokens correspond to latent vectors, not a simple grid.
>
> ---
> >**[Presentation 1]** ... the title term “Identifiable” sounds stronger than the evidence provided in the paper. Furthermore, as there is no experiment on VAE-based or photorealistic RGB tokenizer models,
>
> This sentence is cut off, so we are unclear on the reviewer’s point. The paper provides evidence that ITC positively identifies tokens from the previous frame to copy (“identified token correspondences”), and this identification improves prediction performance.
>
> ---
> >**[Originality 1]** Some supporting elements, such as 3D RoPE, do not seem especially original and should not be counted as major novelty.
>
> 3D RoPE is not the core novelty of the paper, and we do not intend to present it as such. The central contribution is the OT based decoding formulation that explicitly models token correspondence through copy-versus-generate assignments. 3D RoPE is a supporting implementation choice that improves the backbone representation, and we will make that distinction clearer in the revised paper.
>
> ---
> >**[Question 3]** Can the authors better disentangle the roles of OT, binarization, 3D RoPE, and hybrid decoding in the final performance gains?
>
> Based on your suggestion, we disentangle the performance impact of 3D RoPE and OT decoding in the table below. OT makes a significant improvement by itself, beyond the improvement provided by 3D RoPE. We will strengthen and add the table in the revision.
> ||Return (%)|Score (%)|
> |-|-|-|
> |Dedieu et al. (2025)|68.55 ± 0.72|27.24 ± 0.86|
> |+ 3D RoPE|71.85 ± 0.63|33.94 ± 1.10|
> |+ OT (ITC)|72.46 ± 0.45|35.60 ± 0.92|

---

> > ### Author Rebuttal · Reviewer_K2Ga · 2026-04-03
> >
> > Thank you for your rebuttal, I appreciate your responses! However, I think my main concern of how significant the reported results are for general world model is not fully resolved; as pointed out by other reviewers.

---

> > > ### Author Response · Authors · 2026-04-08
> > >
> > > We note that reviewers K2Ga, XqWA, and JWqb raised the same concern about environment diversity; we provide a unified response.
> > >
> > > We agree that adding more environments will make our paper stronger. However, to make a fair comparison with the baselines, we cover all possible visual environments that existing baselines did and ITC shows consistent improvement across various environments compared to the baselines. Note that our paper includes more diverse environments than previous model-based reinforcement learning papers in this area. For example, Dedieu et al. was initially published with only Craftax and MinAtar environments (not Atari 100K). IRIS and DART only evaluate on the Atari 100K environment, and delta-IRIS only evaluates on Crafter and Atari 100K.
> > >
> > > We also want to note that our method and evaluation are not limited to grid-like environments. Using a VQ-VAE tokenizer as we did in Atari 100K, continuous pixel observations turn into latent tokens with spatial relationships, and our method is modelling such latent tokens’ correspondences, which supports all non-grid-like environments.
> > >
> > > We want to share the visualization of some games in Atari 100k, to illustrate that these environments are visually diverse:
> > > - BattleZone (3D-like dynamics as in DOOM): https://www.youtube.com/watch?v=eFqk9tyJ5i4
> > > - PrivateEye (frequent scene changes and sprites’ teleport): https://youtu.be/55S4PJpJ59c?si=o324ZTOo2iX3zZJZ
> > > - ChopperCommand (dynamic movements and camera walks): https://www.youtube.com/watch?v=YBTzXSQQNJU
> > >
> > > In conclusion, we agree that additional environments can be supplementary evidence to prove superior performance of our method, but we believe we already covered possible fair comparisons to baselines and proved superior performance on visually diverse environments.

---

### Official Review · Reviewer_qra1 · 2026-03-12

**Soundness:** 3
**Presentation:** 3
**Significance:** 3
**Originality:** 3
**Overall Recommendation:** 4
**Confidence:** 4

**Summary:**

The paper proposes a method to address next-frame prediction inconsistencies in transformer-based world models. Instead of relying on pure token generation, the authors formulate decoding as an optimal transport problem that assigns each next-frame token to either (i) a copied token from the previous frame or (ii) a newly generated token. This reduces hallucinations and improves long-horizon rollout consistency by enforcing global constraints on token correspondences. Empirically, the method demonstrates strong performance relative to several baselines.

**Compliance With Llm Reviewing Policy:**

Affirmed.

**Final Justification:**

I maintain my rating. The paper addresses a real and important failure mode in transformer-based world models. The core idea is simple, practical motivated and empirically effective. My concerns are reduced but not fully resolved. The authors appropriately agree that the original “structured probabilistic inference” framing is too strong, and the method is better characterised as a structured assignment procedure. The rebuttal also clarifies the block-causal attention setup, resolving my confusion about within-frame autoregressive mismatch. However, the paper still lacks clean ablations isolating key design choices beyond hyperparameter sweeps, and the distance constraint remains a potentially restrictive inductive bias whose failure modes are not systematically analysed. More broadly, the work is most convincing as an effective decoding heuristic with strong empirical value, rather than as a fully understood or theoretically grounded modeling advance.

Overall, I still lean accept because the idea is useful, the empirical improvements are meaningful, and the implementation burden appears low relative to the gains. At the same time, I view this as a borderline accept rather than a strong accept: the contribution is more convincing as a strong engineering improvement than as a fully understood or theoretically grounded advance.

**Key Questions For Authors:**

1. Is OT used only at inference decoding, or does it affect world model training loss?
2. How often does the model copy a token that is wrong but “close enough” under the distance prior?
3. How does ITC behave in environments with larger inter-frame displacement or fast scrolling? Can cd/cost constraints be set without environment knowledge?

**Limitations:**

ITC relies on a distance constraint, which will likely break in games with teleport, large jumps or room changing (likely some Atari games), but the paper does not analyse where this assumption hurts or discuss its limitation.

**Strengths And Weaknesses:**

Strength:
1. Targeting on a long existing problem in the imagination-based world model learning. Hallucinate duplications and disappearance are a known barrier.
2. The method is conceptually clean and sound.
3. The authors have reported empirical performance in different benchmarks and provide a reproduced baseline in Craftax-classic as well as a “baseline with ITC hyperparameters,” which at least acknowledges tuning confounds.

Weakness:
1. The “ structured probabilistic inference problem" framing in the abstract is misleading. It reads like a speculative decoding heuristic.
2. Lack of ablation experiments to justify some design choice like "(i) the distance constraint, (ii) the wildcard penalty term etc".
3. ITC relies on a distance constraint, which will likely break in games with teleport, large jumps or room changing (likely some Atari games), but the paper does not analyse where this assumption hurts or discuss its limitation.

---

> ### Author Rebuttal · Authors · 2026-03-31
>
> Thank you very much for your time and attention in reviewing our paper. We appreciate you highlighting that our targeted problem is long-standing and important, our method is well-conceived and sound, and our empirical performance is comprehensive.
>
> ---
>
>
> > **[Weakness 1]** The “ structured probabilistic inference problem" framing in the abstract is misleading. It reads like a speculative decoding heuristic.
>
> We understand that this phrasing may be ambiguous, so we will update the abstract to use “structured assignment problem” instead of “structured probabilistic inference problem”, which more precisely reflects the role of OT in ITC.
>
> ---
>
>
> > **[Weakness 2]** Lack of ablation experiments to justify some design choice like "(i) the distance constraint, (ii) the wildcard penalty term etc".
>
> We agree that these ablations should be more visible. **Table 9** and **Table 10** in Appendix B contain the ablation experiments for choosing the distance cost and the wildcard penalty term. Note that the distance cost of 0.6 outperforms a cost of 0 (i.e. no distance constraint) in both return and score, justifying the inclusion of a distance constraint. Also note that returns and scores are robust across hyperparameter values. We will summarize these ablations more prominently in the main text.
>
> We do not separately ablate the existence of the wildcard penalty term ($c_w = 0$), because it reduces to the baseline method of always using the generated tokens from the transformer. Without penalizing the generated tokens, the OT step always selects the generated token. Thus, to see the performance of $c_w = 0$, refer to the performance of the underlying baseline *Dedieu et al. (2025)* in **Table 1**.
>
> ---
>
>
> > **[Weakness 3, Limitations]** ITC relies on a distance constraint, which will likely break in games with teleport, large jumps or room changing (likely some Atari games), but the paper does not analyse where this assumption hurts or discuss its limitation.
>
> ITC performs well in Atari games with teleports or room changes, including BattleZone, PrivateEye, and BankHeist. BattleZone takes place in a 3D world with a first-person view, with enemies appearing and disappearing as the player turns left and right, and enemies appearing larger and smaller as the player moves closer and farther. Thus, state prediction in BattleZone cannot be achieved simply by shifting 2D sprites. PrivateEye involves navigating a town across 248 distinct scenes, requiring the agent to handle frequent room changes. BankHeist involves a room change whenever the player moves to the right of the screen to move to the next town. ITC outperforms its baseline *Simulus* in all 3 games as shown in **Table 13** in Appendix E, and additionally outperforms all previous baselines in BattleZone. We will add this discussion explicitly to the revised paper.
>
> From a theoretical perspective, ITC is able to accommodate teleports and room changes when needed because the core of the optimal transport cost matrix is still dependent on the transformer’s predictions ($p_j$ in Equations 1 and 2). The distance constraint is expressed as an added penalty to this cost matrix, but the transformer’s predictions can overcome this distance penalty when the transformer is confident in sudden changes, such as teleports and room changes.
>
> ---
>
>
> > **[Question 1]** Is OT used only at inference decoding, or does it affect world model training loss?
>
> OT is used only at decoding time and does not modify the world-model training objective or loss. It therefore adds no training-loss term and only a negligible inference-time overhead.
>
> ---
>
>
> > **[Question 2]** How often does the model copy a token that is wrong but “close enough” under the distance prior?
>
> Thanks to your question, we performed additional analysis on the accuracy of the copied tokens. Copied tokens have an error rate of 5.79%, while generation for the same token positions have an error rate of 6.76%. This suggests that copying is used selectively and improves accuracy rather than merely choosing a nearby but incorrect token. Also, copying makes the prediction more accurate compared to the original transformer.
>
> ---
>
>
> > **[Question 3]** How does ITC behave in environments with larger inter-frame displacement or fast scrolling? Can cd/cost constraints be set without environment knowledge?
>
> ITC performs well in Atari environments with large interframe displacement and fast scrolling, as discussed above. Cost constraints can be set without environment knowledge, because we did not tune hyperparameters per environment in Atari 100K. Instead, we fixed these hyperparameters across the benchmark, and ITC still improved overall performance. Together with the Appendix B ablations, this suggests that the method is reasonably robust to larger displacement and does not require environment-specific tuning.

---

> > ### Author Rebuttal · Reviewer_qra1 · 2026-04-03
> >
> > Thank you for the rebuttal. I have a follow up question for the block attention you used in the paper. So if you use block attention, so given current frame discrete token $z_t$, how do you generate the next frame token $z_{t+1}$ with block attention? I understand each frame will consist of multiple tokens, say, 16. If you generate these token autoregressively with block attention transformer then there is training and decoding distribution misaligned, right?

---

> > > ### Author Response · Authors · 2026-04-08
> > >
> > > Using block causal attention, we generate multiple tokens in parallel, which makes inference aligned with the training phase. Assuming 16 tokens in a frame, usual causal attention requires 16 inferences to generate the whole frame. Block causal attention requires only 1 inference to generate, since there are no explicit dependencies among the tokens within the frame. We describe it in detail with Figure 7 in Appendix A.
> > >
> > > Let’s say the current frame is represented by 16 discrete tokens $z^1\_t, z^2\_t, z^3\_t, ..., z^{16}\_t$ and we are generating the next frame tokens $z^1\_{t+1}, z^2\_{t+1}, z^3\_{t+1}, ..., z^{16}\_{t+1}$. With standard causal attention, the second token of the next frame $z^2\_{t+1}$ attends to the first token of the next frame $z^1\_{t+1}$, so $z^1\_{t+1}$ must be generated first before $z^2\_{t+1}$. Block causal attention removes the dependency between tokens in the same frame, so $z^2\_{t+1}$ does not attend to $z^1\_{t+1}$ and they can be generated in parallel. That is, $z^2\_{t+1}$ (like all next frame tokens $z^i\_{t+1}$) only attends to the tokens in the previous frame, $z^1\_t, z^2\_t, z^3\_t, ..., z^{16}\_t$. This can be seen in Figure 7 in Appendix A by noting the different output “prediction” tokens shown on the right side of each causal mask. Since block causal attention uses strictly fewer dependencies than standard causal attention, it can be used during both training and inference, so there is no difference or misalignment between training and inference.

---

### Decision · Program_Chairs · 2026-04-30

**Decision:**

Accept (regular)

**Comment:**

The paper received mixed but overall positive reviews. Reviewers generally agreed that it addresses an important failure mode in transformer-based world models, and that the proposed OT-based copy-or-generate decoding mechanism is intuitive, technically sound, and practically useful. The experiments show consistent improvements over direct baselines, and the rebuttal clarified key points regarding the decoding-time role of OT, fairness of comparison, and computational overhead.

While some reviewers raised concerns about broader applicability, I find the evaluation sufficient for the paper's scope. The method is tested on several commonly used benchmarks in prior work, including Craftax, MinAtar, and Atari 100K, and demonstrates meaningful gains in these settings. The remaining concerns are mainly about framing, presentation, and the need for clearer ablations, rather than the core contribution itself. Overall, I believe this is a solid and useful contribution that merits acceptance.